# The mlpt/Ubr3/Svb module comprises an ancient developmental switch for embryonic patterning

Suparna Ray[1†], Miriam I Rosenberg[2†*], Hélène Chanut-Delalande[3], Amélie Decaras[4], Barbara Schwertner[1], William Toubiana[4], Tzach Auman[2], Irene Schnellhammer[1], Matthias Teuscher[1‡], Philippe Valenti[3], Abderrahman Khila[4], Martin Klingler[1], François Payre[3*]

[1]Department of Biology, Developmental Biology, University of Erlangen-Nuremberg, Erlangen, Germany; [2]Department of Ecology, Evolution and Behavior, Hebrew University of Jerusalem, Jerusalem, Israel; [3]Centre de Biologie du Développement, Université Paul Sabatier de Toulouse, Toulouse, France; [4]Institut de Génomique Fonctionelle de Lyon, Lyon, France

**\*For correspondence:**
miriam.rosenberg@nyu.edu (MIR);
francois.payre@univ-tlse3.fr (FP)

[†]These authors contributed equally to this work

**Present address:** [‡]Institut für Zoologie, Abteilung für Entwicklungsbiologie, Universität zu Köln, Cologne, Germany

**Competing interests:** The authors declare that no competing interests exist.

**Abstract** Small open reading frames (smORFs) encoding 'micropeptides' exhibit remarkable evolutionary complexity. Conserved peptides encoded by *mille-pattes (mlpt)/polished rice (pri)/tarsal less (tal)* are essential for embryo segmentation in *Tribolium* but, in *Drosophila*, function in terminal epidermal differentiation and patterning of adult legs. Here, we show that a molecular complex identified in *Drosophila* epidermal differentiation, comprising Mlpt peptides, ubiquitin-ligase Ubr3 and transcription factor Shavenbaby (Svb), represents an ancient developmental module required for early insect embryo patterning. We find that loss of segmentation function for this module in flies evolved concomitantly with restriction of Svb expression in early *Drosophila* embryos. Consistent with this observation, artificially restoring early Svb expression in flies causes segmentation defects that depend on *mlpt* function, demonstrating enduring potency of an ancestral developmental switch despite evolving embryonic patterning modes. These results highlight the evolutionary plasticity of conserved molecular complexes under the constraints of essential genetic networks.

**Editorial note:** This article has been through an editorial process in which the authors decide how to respond to the issues raised during peer review. The Reviewing Editor's assessment is that all the issues have been addressed (see decision letter).

DOI: https://doi.org/10.7554/eLife.39748.001

## Introduction

Animal genomes transcribe a variety of long-non-coding RNAs, whose functions are not yet fully understood (*Cech and Steitz, 2014*; *Guttman and Rinn, 2012*; *Perry and Ulitsky, 2016*). A large body of evidence increasingly supports translation of so called 'micropeptides' from small open reading frames < 100 amino acids (also called small ORFs, smORFs or sORFs) encoded in long 'non-coding' RNAs (*Couso and Patraquim, 2017*; *Plaza et al., 2017*). Owing to their relatively recent discovery and experimental validation, micropeptides represent an overlooked reservoir of evolutionary and regulatory material. Identification of their developmental functions has hitherto been limited to a handful of cases and their putative contribution to animal evolution is unknown.

One of the best-known cases of smORF-encoded peptides called *mille-pattes/tarsal less/polished rice* (10 to 32 amino acids; hereafter referred to as *mlpt*), are conserved across arthropods, a taxon representing over 400 million years of evolutionary time (*Galindo et al., 2007*; *Kondo et al., 2007*;

*Savard et al., 2006*). It has been shown that *Drosophila* embryos lacking *mlpt* function develop with proper segment patterning, but exhibit strong defects in epidermal differentiation, notably the absence of cuticular trichomes (*Galindo et al., 2007*; *Kondo et al., 2007*). In the fly epidermis, Mlpt peptides act through post-translational control of Ovo/Shavenbaby (Svb)(*Kondo et al., 2010*), a transcription factor well-established as the key regulator of trichomes (*Payre et al., 1999*). Svb is translated as a transcriptional repressor (*Kondo et al., 2010*) and Mlpt peptides bind to and activate an E3 ubiquitin ligase, Ubr3, enabling its interaction with Svb (*Zanet et al., 2015*). Formation of the Mlpt/Ubr3/Svb complex leads to proteasome degradation of the Svb N-terminal repression domain thereby, releasing a shorter Svb protein that functions as a transcriptional activator (*Kondo et al., 2010*; *Zanet et al., 2015*). Upon processing, Svb activates the expression of cellular effectors (*Chanut-Delalande et al., 2006*; *Fernandes et al., 2010*; *Menoret et al., 2013*), comprising a gene network deeply conserved throughout arthropods (*Chanut-Delalande et al., 2006*; *Li et al., 2016*; *Spanier et al., 2017*). Hence, a central function of Mlpt peptides during *Drosophila* development is to provide temporal control of Svb transcriptional activity, exemplified by their role in epidermal differentiation (*Chanut-Delalande et al., 2014*; *Zanet et al., 2016*).

Independently, *Savard et al. (2006)* discovered an essential function for this locus in the formation of abdominal segments in the flour beetle, *Tribolium castaneum* (*Savard et al., 2006*). In beetles, RNAi knockdown of *mlpt* caused posterior truncation of the embryo, with a loss of abdominal segments, as well as the transformation of remaining anterior abdominal segments to thoracic fate, leading to a distinctive phenotype of extra pairs of legs (mille-pattes is French for centipede). Additional work established that *mlpt* acts as a gap gene in *Tribolium* (*Boos et al., 2018*; *Ribeiro et al., 2017*; *Savard et al., 2006*; *van der Zee et al., 2006*; *Zhu et al., 2017*), where more limited homeotic transformations often accompany loss of gap gene function (*Bucher and Klingler, 2004*; *Cerny et al., 2005*; *Marques-Souza et al., 2008*). Unlike *Drosophila* which has evolved a derived mode of segmentation (called 'long germ') in which all segments are formed nearly simultaneously in the syncytial environment of the blastoderm, *Tribolium* is more representative of the ancestral mode of segmentation in insects (*Peel et al., 2005*). Most insects, like beetles, develop as short/intermediate germband embryos where only head and thorax are patterned in the blastoderm, whereas most or all posterior segments are added from a posterior 'growth zone' (*Davis and Patel, 2002*; *Liu and Kaufman, 2005*; *Rosenberg et al., 2009*). In spite of the striking absence of embryonic patterning defects in *Drosophila mlpt* mutants, the strong phenotype of *mlpt* in beetles suggested an ancestral function of the peptides in segmentation, a hypothesis we set out to investigate through their functional analysis across insect species.

## Results

### Identification of *mlpt* partners Svb and Ubr3 in *Tribolium* segmentation

We sought to identify functional partners for Mlpt peptides that explain their function in *Tribolium* segmentation. The genome-wide iBeetle RNAi screen in *Tribolium* (*Dönitz et al., 2018*; *Dönitz et al., 2015*; *Schmitt-Engel et al., 2015*) allowed a large-scale search for patterning genes leading to a *mlpt*-like mutant phenotype, as a means of identifying candidate partners.

Knockdown of >5000 genes revealed only a few candidates sharing such a segmentation phenotype (*Supplementary file 1A*). Further analyses validated a gene producing a reproducible phenotype that is highly similar to that of *mlpt*. Unexpectedly, this candidate was *Tc-ubr3,* the E3 ubiquitin ligase now known to be the molecular target of Mlpt peptides for epidermal differentiation in flies. In *Tribolium*, the *Tc-ubr3* RNAi phenocopies *mlpt* RNAi with severely shortened larvae due to the absence of many abdominal segments as well as telson appendages (*Figure 1A–C* and *Figure 1— figure supplement 1*). Furthermore, as in *mlpt* RNAi, the remaining 'abdominal' segments appear to be transformed to a thoracic fate since they bear extra legs and often spiracles resembling those present on the second thoracic segment (*Figure 1A–C* and *Figure 1—figure supplements 1* and *2*). The *Tc-Ubr3* phenotype can exceed *mlpt* RNAi in severity, with strongly affected legs developing shorter and poorly differentiated segments (*Figure 1F,G* and *Figure 1—figure supplements 1* and *2*). However, the overall similarity between *mlpt* and *Tc-Ubr3* phenotypes (*Table 1*) suggested that the complete fly epidermal module may be conserved for *Tribolium* segmentation.

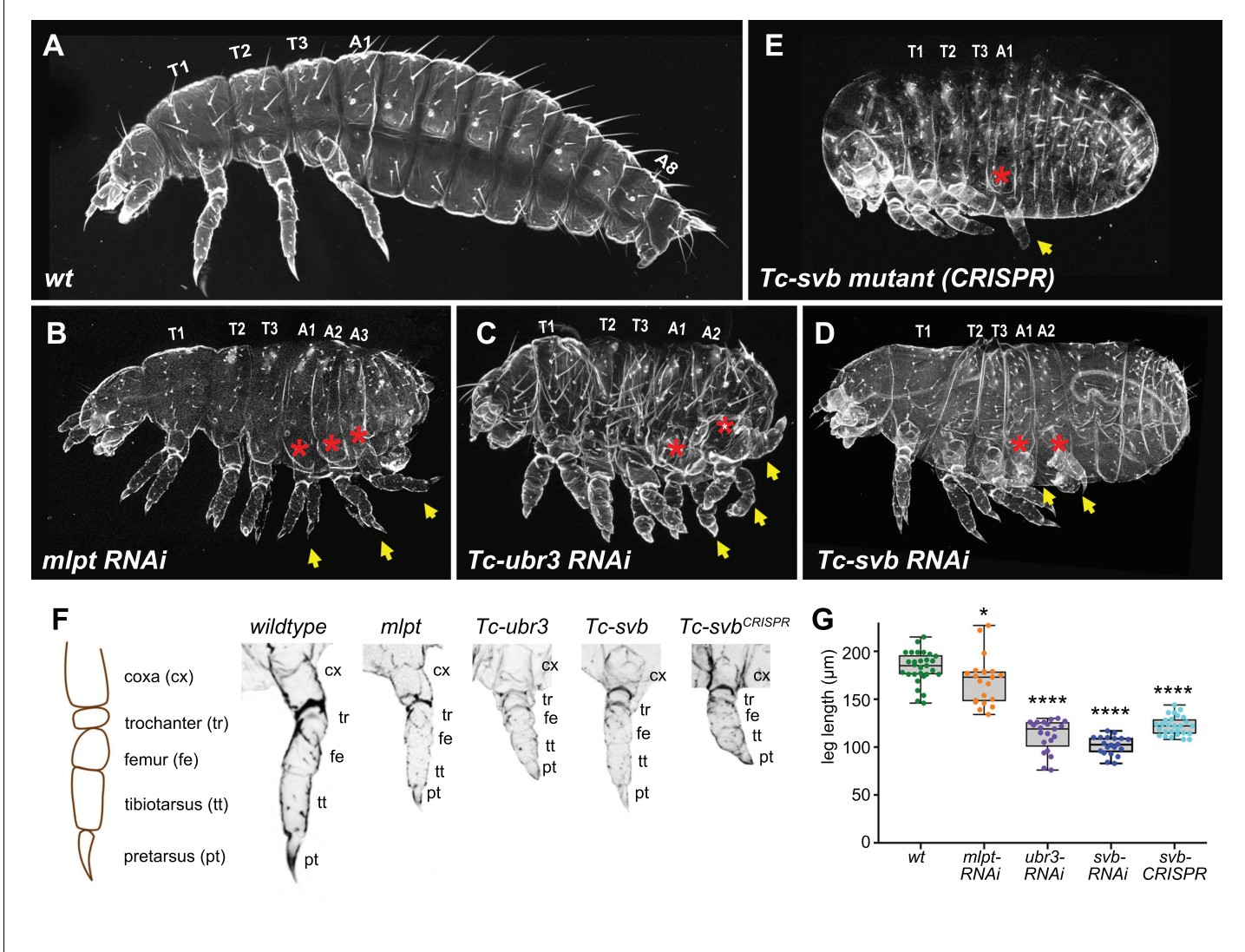

**Figure 1.** A cooperative segmentation function of the module *Mlpt/Svb/Ubr3* in *Tribolium*. Cuticle phenotypes of *Tribolium* first instar larvae from following genotypes: *wild type* (**A**), *mlpt* RNAi (**B**), *Tc-ubr3* RNAi (**C**), *Tc-svb* RNAi (**D**), and *Tc-svb* CRISPR mutant (**E**). Depletion of *mlpt*, *Tc-svb*, and *Tc-ubr3* causes highly similar segmentation phenotypes, characterized by a reduction in segment number, the presence of extra-legs (arrows) suggestive of transformation of abdominal segments towards a thoracic fate (red asterisks), and the frequent absence of terminal structures. (**F**) Knockdown of each of the three genes leads to shortened 'true-thoracic' legs, with rounded and often poorly separated distal segments. The scheme represents a larval leg with corresponding segments; pictures portray an example of prothoracic leg (**T1**) in *wildtype*, *mlpt*, *Tc-ubr3* and *Tc-svb* inactivation. (**G**) Quantification of the reduction in leg length, estimated by the distance between coxa/trochanter boundary to the pretarsus tip. Data were analyzed by one-way ANOVA using multiple comparison tests against wild-type values. *, p-value<0,05; ****, p-value<0,0001. Source data for *Figure 1G* are found in Source Data File 1.

DOI: https://doi.org/10.7554/eLife.39748.002

The following figure supplements are available for figure 1:

**Figure supplement 1.** Cuticle of *Tribolium* larvae showing different examples of *Tc-ubr3* RNAi phenotypes.
DOI: https://doi.org/10.7554/eLife.39748.003

**Figure supplement 2.** Extreme phenotypes of *Tribolium mlpt* and *Tc-ubr3* knockdowns.
DOI: https://doi.org/10.7554/eLife.39748.004

**Figure supplement 3.** *Tribolium Tc-svb* RNAi larval cuticles of increasing phenotypic strength.
DOI: https://doi.org/10.7554/eLife.39748.005

**Figure supplement 4.** Schematic representation of *Tc-shavenbaby* locus (**A**) and transcript (**B**), showing the site at which a GFP-containing marker plasmid was inserted by CRISPR/cas9 genome editing (see also Materials and methods).
DOI: https://doi.org/10.7554/eLife.39748.006

*Figure 1 continued*

**Figure supplement 5.** Shavenbaby protein features are conserved across insect species.
DOI: https://doi.org/10.7554/eLife.39748.007

In support of this hypothesis, we found that RNAi knockdown of *Tc-svb* also leads to a highly penetrant abdominal truncation and homeotic transformation phenotype that resembles that of *mlpt* and *Tc-ubr3* knockdowns (*Figure 1D* and *Figure 1—figure supplement 3*). Knockdown *Tc-svb* larvae are characterized by the presence of legs on the first two 'abdominal' segments, even in the weaker segmentation phenotypes, wherein legs on segment 'A1' are often reduced to mere stumps (*Figure 1D* and *Figure 1—figure supplement 3*). Presence of T2-like spiracles on 'A1' and the absence of spiracles on 'A2' in *Tc-svb* knockdowns suggest their transformation into thoracic segments, T2 and T3, respectively. In the stronger phenotypes, the body (including the head) is very compact and the posterior abdominal segments are fused (*Figure 1D* and *Figure 1—figure supplement 3*). Although the extent of abdominal segment loss is weaker than for *mlpt* and *Tc-ubr3* RNAi, all *Tc-svb* RNAi larvae are clearly shortened compared to the wild type. As with *mlpt* and *Tc-ubr3* knockdown, leg segments are severely shortened and rounded, and pretarsi are reduced in *Tc-svb* knockdowns (*Figure 1F,G* and *Figure 1—figure supplement 3*).

In summary, in spite of some phenotypic differences, *Tc-ubr3*, *mlpt*, and *Tc-svb* larvae share several critical similarities, including some degree of posterior truncation, transformation of remaining abdominal segments towards thoracic identity, shortened leg segments with a 'bubble-like' terminus, and missing telson appendages (*Table 1*). The fact that the three functional partners identified in the fly epidermis share similar phenotypes in beetle embryonic patterning led us to hypothesize that they may act as a functional module for control of *Tribolium* segmentation. We accumulated several lines of evidence that support this view.

First, we generated a *Tc-svb* mutant using CRISPR/cas9 genome editing (see Materials and methods). Molecular characterization of the *Tc-svb* locus in wild-type and CRISPR-mutants indicated that this allele was a strong hypomorph, if not a null (*Figure 1—figure supplement 4*). CRISPR knockout of *Tc-svb* phenocopies the observed RNAi defects (*Figure 1A–E*), and highlights an additional phenotype consisting of a considerable thinning of the epidermal cuticle, similar to what has been observed in the fly (*Andrew and Baker, 2008*). As in *Tc-svb* RNAi, ectopic legs or leg rudiments are present on A1 and A2. Additional phenotypes observed in mutants include shorter and misdifferentiated legs (*Figure 1A,E–G*).

Second, if Tc-Svb functions molecularly via the Mlpt/Ubr3 complex, it should bear the same characteristic protein features. We therefore compared the sequence and predicted characteristics of the Tc-Svb protein to that of the fly protein (*Figure 1—figure supplement 5*). In flies, limitation of Ubr3-mediated proteasome degradation to the N-terminal domain of Svb has been linked to intrinsically disordered disposition of this region (*Zanet et al., 2015*), as opposed to the C-terminal transactivation and DNA-binding domains that resist proteasome degradation. Despite rapid evolution of Svb protein sequence outside the zinc-finger region (*Kumar et al., 2012*), this predicted disordered

**Table 1.** Summary of *Tribolium* phenotypes resulting from RNAi-mediated depletion of *mlpt*, *Tc-Ubr3*, *Tc-Svb*, as well as those observed in *Tc-Svb CRISPR* mutants.

In each case, a total of 20 animals were scored. Data show the average number of deleted abdominal segments, missing terminal appendages (urogomphi) and number of pairs of extra legs. Cuticle defects were scored as normal-looking (-), mild (+) and strong (+++) thinning. For leg length, the distance from coxa/trochanter joint to leg tip (see *Figure 1*) was measured in segment T3.

| | Deleted abdominal segments | Urogomphi missing | Thoracic leg length ($\mu$m) | Extra legs | Cuticle thinning |
|---|---|---|---|---|---|
| *Wild type* | 0 | 0 | 183 | 0 | - |
| *mlpt-RNAi* | 3.8 | 2 | 170 | 4.3 | - |
| *Tc-ubr3 RNAI* | 5.1 | 2 | 112 | 3.9 | + |
| *Tc-svb RNAi* | 0.5 | 1.5 | 102 | 3.2 | + |
| *Tc-svb CRISPR* | 1.0 | 1.7 | 122 | 1.65 | +++ |

DOI: https://doi.org/10.7554/eLife.39748.008

disposition pattern remains strikingly conserved for Svb in *Tribolium* and other insects (*Figure 1—figure supplement 5A–C*). Tc-Svb also displays strong conservation of the protein motifs identified in flies as required for Svb processing: the maturation site (*Kondo et al., 2010*) and the N-terminal region (*Figure 1—figure supplement 5D–F*) bound and ubiquitinated by Ubr3 to target Svb to the proteasome (*Zanet et al., 2015*). Indeed, other top hits detected by the iBeetle screen correspond to factors involved in ubiquitin proteasome degradation (*Supplementary file 1A*).

Third, we examined mRNA expression of all three components during *Tribolium* embryogenesis. As in flies, *Tc-Ubr3* is expressed ubiquitously in the beetle embryo, as expected for an enzyme with additional widespread functions, including in DNA repair (*Meisenberg et al., 2012*) and apoptosis (*Huang et al., 2014*). In contrast, *Tc-svb* and *mlpt* display a dynamic pattern during both blastoderm and germband stages of *Tribolium* embryogenesis (*Figure 2* and *Figure 2—figure supplement 1*). Importantly, *Tc-svb* is co-expressed with *mlpt* within the pre-growth zone at the onset of gastrulation (*Figure 2B,B'*). The posterior *Tc-svb* domain evolves into a strong anterior band flanking the serosa and a more diffuse posterior expression (*Figure 2C'*), while *mlpt* has much stronger posterior expression (*Figure 2C*). As the embryo extends, *Tc-svb* forms two distinct expression domains flanking the strong *mlpt* expression domain (*Figure 2D,D'*), suggesting that high levels of *mlpt* and *Tc-svb* expression may be mutually repressive (*Figure 2—figure supplement 2*). Subsequently, *Tc-svb* and *mlpt* expression domains shift, wave-like, anteriorly, while anterior *Tc-svb* expression fades and its posterior expression detaches from the posterior end (*Figure 2E,E'*). The interaction at such interfaces of the complementary domains may be critical for patterning of the abdominal segments.

The co-expression of *mlpt* and *Tc-svb* in the posterior growth zone helps explain why they share similar segmentation phenotypes. Examination of the segmental marker *wg* confirms that abdominal segments are specifically disrupted in *mlpt, Tc-svb*, and *Tc-ubr3* RNAi embryos, while thoracic segments are formed normally (*Figure 2F–I*). This is of interest since in the short germ embryo of

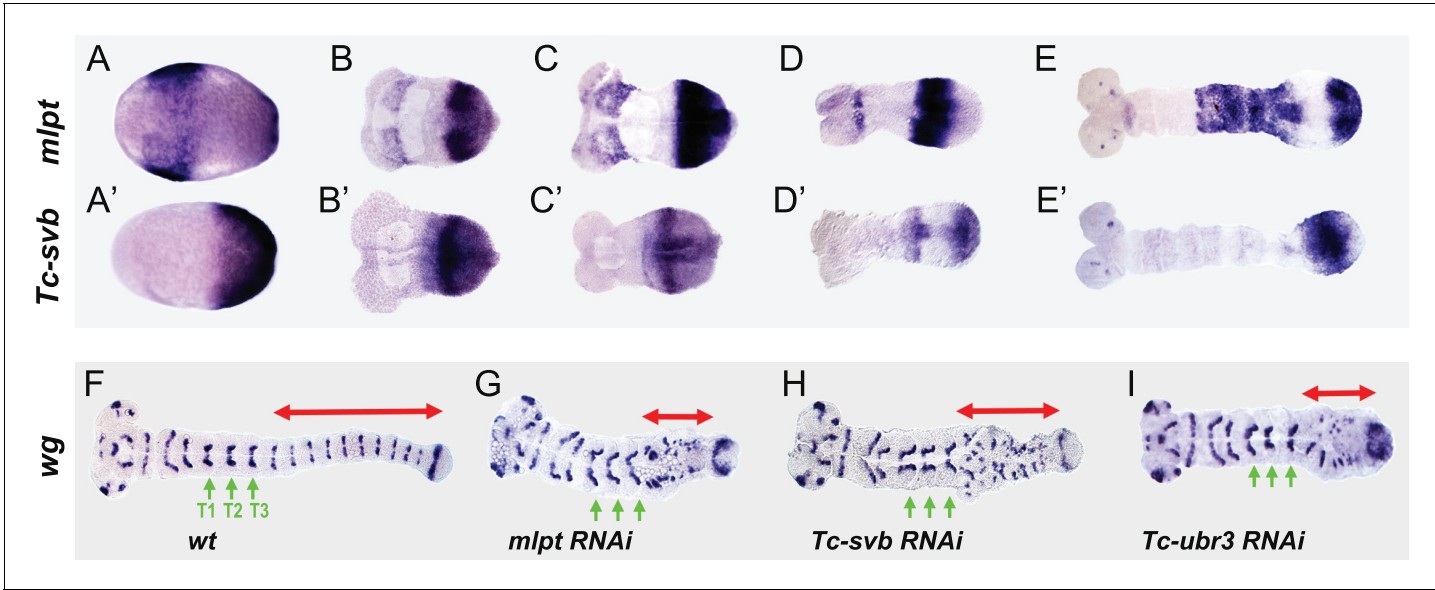

**Figure 2.** Tc-mlpt and Tc-Svb embryonic expression and function of the mlpt/svb/ubr3 module in abdominal patterning. (A–E') Whole mount in-situ hybridization of *Tribolium* embryo showing mRNA expression of *mlpt* and *Tc-svb* from late blastoderm (A,A') through extending germband stages (B, B', C,C', D,D', E,E'), highlighting their complementary expression pattern (F–I) Wingless (*wg*) expression in *wild type* (F), *mlpt*-RNAi (G), *Tc-svb*- RNAi (H) and Tc-*ubr3*- RNAi (I) *Tribolium* embryos. Abdominal segments are highlighted with red arrow. In all three knockdown conditions, *wg* segmental stripes are disrupted right after the last (T3) thoracic stripe. Thoracic segments (T1–T3) are indicated by green arrows.
DOI: https://doi.org/10.7554/eLife.39748.009

The following figure supplements are available for figure 2:

**Figure supplement 1.** Embryonic expression of *Tribolium svb*.
DOI: https://doi.org/10.7554/eLife.39748.010

**Figure supplement 2.** *mlpt RNAi* causes de-repression of *svb* expression in short germ embryos of *Tribolium* and *Oncopeltus*.
DOI: https://doi.org/10.7554/eLife.39748.011

*Tribolium*, the head and the first thoracic segment form in the syncytial blastoderm, while after cellularization, subsequent segments continue to arise in a sequential manner from the posterior growth zone (*Liu and Kaufman, 2005*; *Rosenberg et al., 2009*).

In summary, patterns of *mlpt* and *Tc-svb* expression during *Tribolium* embryonic development are often complementary, and at times, overlapping. Loss of function phenotypes of *mlpt, Tc-svb* and *Tc-ubr3* suggest that a functional module for *mlpt* discovered in *Drosophila* trichome patterning also works in concert in embryonic segmentation, leg patterning and cuticle formation in *Tribolium*.

## Complementarity of expression of *mlpt* and *svb* is deeply conserved in insects

Our data revealed a surprising and essential role for this gene module in controlling posterior segment formation and identity in *Tribolium*. To determine whether this tripartite module may function in embryonic development of other insects, we investigated the expression patterns of *mlpt, ubr3* and *svb* in additional, more basal insect species: the water strider, *Gerris buenoi* (*Gb*; Hemiptera, Gerridae) and the milkweed bug, *Oncopeltus fasciatus* (*Of*; Hemiptera, Lygaeidae).

*Figure 3* highlights the expression patterns of these genes throughout embryogenesis. The early development of the milkweed bug and the water strider are quite similar. *Ubr3* expression is ubiquitous in both *Oncopeltus* and *Gerris* and was not examined further. *mlpt* and *svb* expression in the early hemipteran embryo are observed in strong domains at the anterior of the blastoderm embryo (*e.g., Oncopeltus, Figure 3A,A'*), with additional posterior *Of-svb* expression at the future site of invagination which becomes broad expression throughout the early growth zone (*Figure 3A* and *Figure 3—figure supplement 1*). This pattern persists, until a transition to a transient overlap in the early growth zone (*Figure 3—figure supplement 1*). Subsequently, expression of *svb* and *mlpt* resolve into complementary /overlapping domains within the growth zone (*Figure 3B–E'* and *Figure 3—figure supplement 1*). *Of-mlpt* expression is also diffusely expressed through recently added segments anterior to the growth zone (*Figure 3C'*). Later expression in both species is seen in presumptive neurons in the central nervous system, as well as in the limb buds and mouth parts (*Figure 3C–F'* and *Figure 3—figure supplement 1*), consistent with a function in patterning the leg and head appendages.

These data hint at a surprising role for this gene module in controlling segment formation and identity in representatives of the Coleoptera and Hemiptera, but not Diptera.

## Conserved function of *mlpt/ubr3/svb* gene module in insect segmentation

We next tested whether and how broadly *mlpt, svb*, and *ubr3* may functionally cooperate during embryogenesis in these additional short germ insects. RNAi against each of these genes caused severe segmentation and patterning defects both in *Gerris* and *Oncopeltus*.

Embryos of hemimetabolous insects, including water striders and milkweed bugs, complete embryogenesis and undergo a series of molts through which they reach adulthood. These intermediate nymph stages or hatchlings exhibit the full complexity of adult structures. In *Gerris* and *Oncopeltus*, the wild type hatchling possesses three long pairs of legs, which extend along the ventral side, curling around the posterior, as well as a long pair of antennae that extend posteriorly along the ventral midline (*Figure 4A,A'; E,E'*). *mlpt* RNAi in both *Gerris* and *Oncopeltus* resulted in the loss of posterior abdominal segments and fusion of thoracic segments, with shortened rounded legs that terminate proximal to the trunk; reduction and fusion of head appendages is also apparent ( and *Figure 4—figure supplement 1,2*). In *Oncopeltus*, severely affected embryos fail to gastrulate, resulting in an everted gut (*Figure 4—figure supplement 1A",B"*). *Gb-* and *Of-svb* RNAi also resulted in the loss of abdominal segments and rounding of more distally truncated legs (*Figure 4C, C'; G,G'*). Following *Gb-svb* RNAi, even mildly affected prenymphs exhibited significant reduction in leg length (*Figure 4—figure supplement 3*). Examination of molecular markers confirmed strong defects in embryonic segmentation and appendage formation in both *Gerris* (*Figure 4— figure supplement 4,5*) and *Oncopeltus* (*Figure 4—figure supplement 6*). *ubr3* RNAi in both species gave the most severe phenotype, reflecting its presumed additional functions independent of *svb* and *mlpt* (*Figure 4D,D'; H,H'*). In *Oncopeltus*, severe *ubr3* RNAi embryos were almost completely ablated, leaving unidentifiable ectodermal tissue connected to everted presumptive visceral tissue (*Figure 4—*

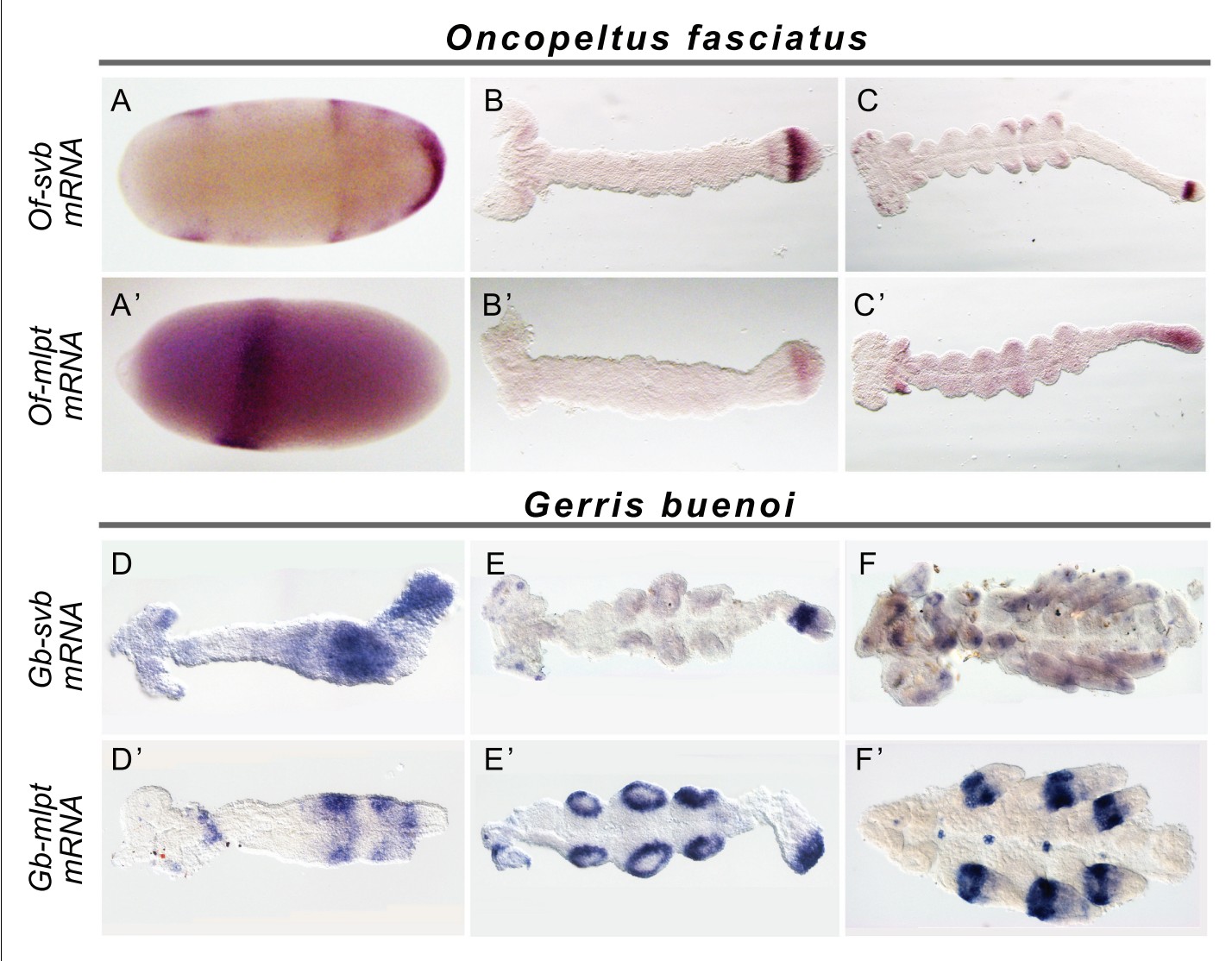

**Figure 3.** Expression of *mlpt* and *svb* in hemipteran embryos. Whole mount in situ hybridization of *svb* and *mlpt* mRNA in *Oncopeltus* (**A–C**) and *Gerris* (**D–F**) embryos at early, mid-germ and late embryonic stages. (**A–C**) Oncopeltus embryonic expression. At early stages, *Of-svb* expression is mainly expressed in two domains (anterior head and thoracic segments) (**A**) *Of-mlpt* is restricted to a single strong stripe in presumptive head segments (**A'**). Then, *Of-svb* is expressed faintly in the head lobes and strongly in two growth zone stripes (**B**) while *Of-mlpt* is exclusively expressed in the posterior of the growth zone (**B'**). Late embryos express *Of-svb* expression in a strong stripe in the middle of the growth zone, as well as in putative head neurons and limb buds (**C**). At this stage, faint *Of-mlpt* mRNA expression is detected in the head appendages, putative head and thoracic segments, and strong but diffuse expression throughout the growth zone (**C'**). (**D–F**) Gerris embryonic expression. In early embryos, *Gb-svb* is faintly expressed in the head and thorax, with stronger expression in the abdomen of the early germ band (**D**), when *Gb-mlpt* expression is restricted to a thoracic stripe and two distinct abdominal domains, abutting *Gb-svb* expression (**D'**). Mid germ band embryos have more restricted *Gb-svb* expression, in a stripe in the growth zone, in putative neurons in the head, and faintly in limb buds (**E**) while they exhibit strong expression of *Gb-mlpt* in the limb buds, and in the posterior of the growth zone, immediately adjacent to strong *Gb-svb* expression. Late stage embryos exhibit faint banded expression of *Gb-svb* in the legs and head appendages, and in foci in the head (**F**) whereas they exhibit strong *Gb-svb* expression in the mature limbs, and in foci of expression along the embryo midline (**F'**).

DOI: https://doi.org/10.7554/eLife.39748.012

The following figure supplement is available for figure 3:

**Figure supplement 1.** Expression of *Of-mlpt* and *Of-svb* mRNA throughout *Oncopeltus* embryogenesis.

DOI: https://doi.org/10.7554/eLife.39748.013

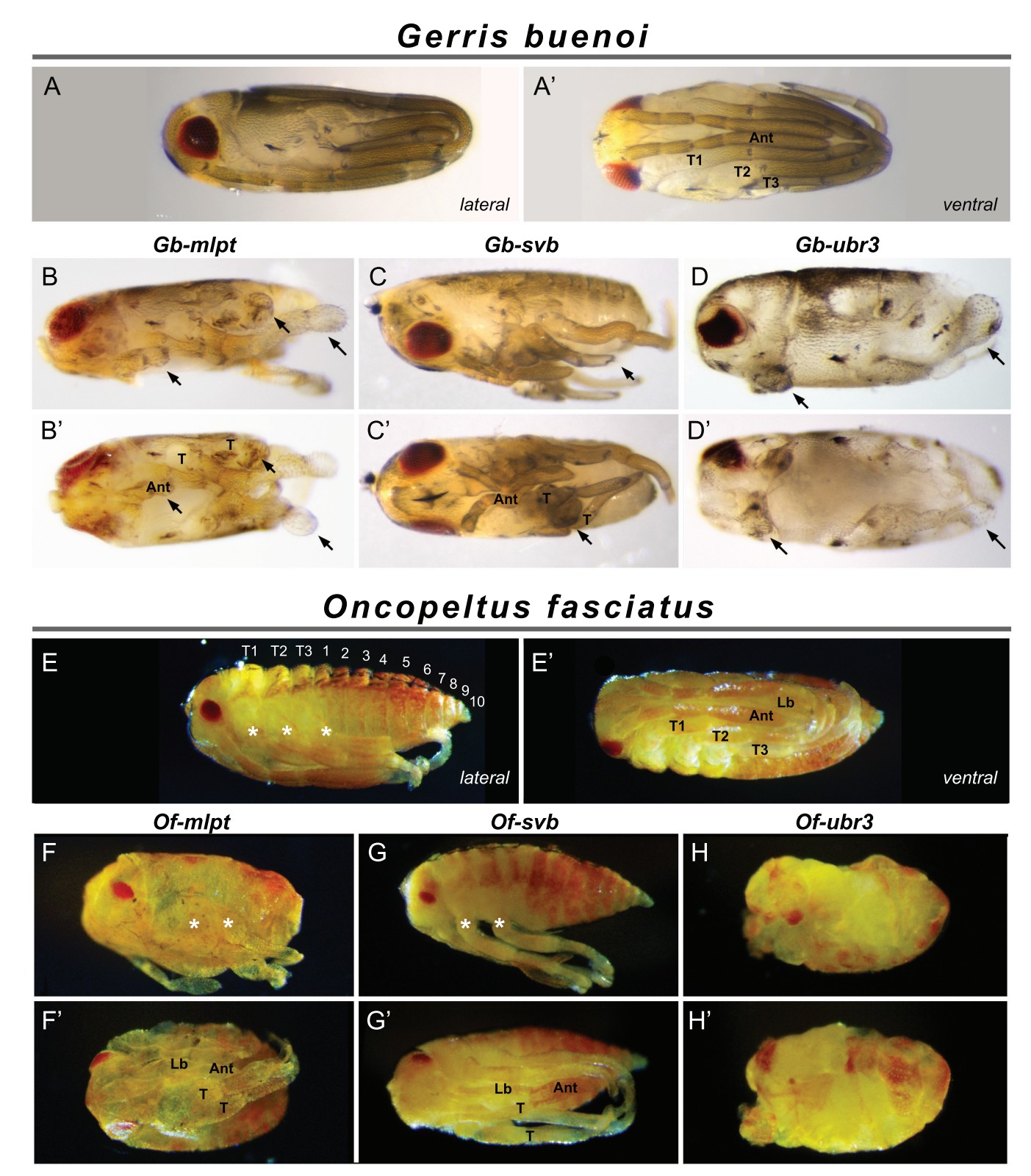

**Figure 4.** Knockdown of *mlpt*, *svb*, and *ubr3* affects embryo segmentation in *Gerris* (A–D') and *Oncopeltus* (E–H'). Hatchlings are presented in lateral (A–D, E–H) and ventral (A'–D' and E'–H') views. Wild type *Gerris* pre-nymphs possess red pigmented eyes, and antennae that extend along the ventral side of the embryo, terminating between long legs which wrap around the embryo (A–A'). Both *Gb-mlpt* and *Gb-svb* RNAi embryos display posterior truncation, as well as loss and/or fusion of legs and head appendages (B–C'). *Gb-mlpt* embryos show altered eye morphology. *Gb-ubr3* embryos

*Figure 4 continued on next page*

*Figure 4 continued*

exhibit more severe posterior, leg and eye phenotypes (D,D'). (E–H') Phenotypes of *wild type Oncopeltus* (E–E') hatchlings alongside *Of-mlpt* (F–F'), *Of-svb* (G–G) and *Of-ubr3* (H–H') RNAi. *Of-mlpt* and *Of-svb* RNAi causes posterior truncation, with the fusion/loss of thoracic segments, shortened legs and head appendages, and a reduced eye. *Of-ubr3* RNAi displays similar phenotypes but stronger than *Of-mlpt* and *Of-svb* RNAi, with an apparent loss of axial polarity in severely affected *Of-ubr3* RNAi embryos. Source data for *Figure 4—figure supplements 1–3* are found in *Source Data 1*.
DOI: https://doi.org/10.7554/eLife.39748.014

The following figure supplements are available for figure 4:

**Figure supplement 1.** Phenotypes of increasing strength for *Of-mlpt*, *Of-svb* and *Of-ubr3* RNAi in *Oncopeltus*.
DOI: https://doi.org/10.7554/eLife.39748.015
**Figure supplement 2.** Effects of *Gb-mlpt*, *Gb-svb*, *Gb-ubr3* RNAi depletion in *Gerris*.
DOI: https://doi.org/10.7554/eLife.39748.016
**Figure supplement 3.** *Gb-svb* RNAi treatment induces defects in *Gerris buenoi* leg development and differentiation.
DOI: https://doi.org/10.7554/eLife.39748.017
**Figure supplement 4.** *Gb-svb* RNAi *Gerris* embryos display developmental defects.
DOI: https://doi.org/10.7554/eLife.39748.018
**Figure supplement 5.** Effects of *Gb-svb*-RNAi and *Gb-mlpt* RNAI treatment in *Gerris* appendages.
DOI: https://doi.org/10.7554/eLife.39748.019
**Figure supplement 6.** *Of-svb* and *Of-mlpt* RNAi leads to defects in embryonic segmentation.
DOI: https://doi.org/10.7554/eLife.39748.020
**Figure supplement 7.** Cuticle defects in hemipteran embryos depleted for *svb*, *mlpt* and *ubr3*.
DOI: https://doi.org/10.7554/eLife.39748.021

*figure supplement 1*). More mildly affected embryos showed some apparent segment identity, with head and eyes, but no appendages and limited evidence for correct axial polarity (*Figure 4H,H'* and *Figure 4—figure supplement 1*). As observed in *Tribolium*, RNAi, knockdown of *mlpt*, *svb*, and *ubr3* in hemiptera also leads to strong cuticle defects including the loss of trichomes (*Figure 4—figure supplement 7*).

Taken together, these data highlight deep conservation of the Mlpt/Ubr3/Svb module in basal, 'short germ' insects, both in patterns of embryonic expression and in segmentation function.

## Functional conservation of Mlpt/Ubr3/Svb module in alternative long-germ insects

Since all basally branching insect species examined showed evidence of conserved function of this module in segmentation, we assayed the expression and putative function of the tripartite gene module in the jewel wasp *Nasonia vitripennis*, an insect species with a derived segmentation mode.

Like *Drosophila*, *Nasonia* has evolved long germ embryogenesis, in which the embryo is mostly patterned in the context of the syncytial blastoderm, and which has evolved independently several times in the insect phylum (*Liu and Kaufman, 2005*; *Misof et al., 2014*; *Rosenberg et al., 2009*). Previous work has identified the key role of maternal determinants and gap genes in *Nasonia*, which largely resemble that of *Drosophila* where most segmentation occurs in the blastoderm (*Brent et al., 2007*; *Lynch et al., 2006*), with some residual character of delayed segment patterning of the most posterior segments after cellularization (*Rosenberg et al., 2009*).

In *Drosophila*, whereas *svb* early expression is absent from posterior segments and restricted to two stripes in the head (*Mével-Ninio et al., 1995*) (*Figure 5A,B*), *tal/mlpt* is expressed more broadly throughout the blastoderm (*Figure 5D,E*) with a striped pattern evoking that of the pair-rule gene *hairy* (*Galindo et al., 2007*). Consistent with previous studies, we confirmed that the absence of *tal/mlpt*, *svb* or *Ubr3* does not alter segmentation, as deduced from analysis of mutant embryos lacking both maternal and zygotic contribution for each of the three genes (*Figure 5M–P* and *Figure 5—figure supplement 1*).

In contrast, in *Nasonia*, both *Nv-mlpt* and *Nv-svb* are expressed in the early embryo, in adjacent prominent stripes at the posterior region of embryo (*Figure 5G–K*) that acts as the progenitor of the late-forming segments (*Rosenberg et al., 2014*). *Nv-svb* is also expressed in a prominent stripe in the middle of the embryo (*Figure 5G–H* and *Figure 5—figure supplement 2*), similar to expression of the thoracic gap gene, *Nv-krüppel* (*Brent et al., 2007*), while *Nv-mlpt* expression has an anterior cap, and broad expression posterior to the *Nv-svb* domain (*Figure 5J–K* and *Figure 5—figure*

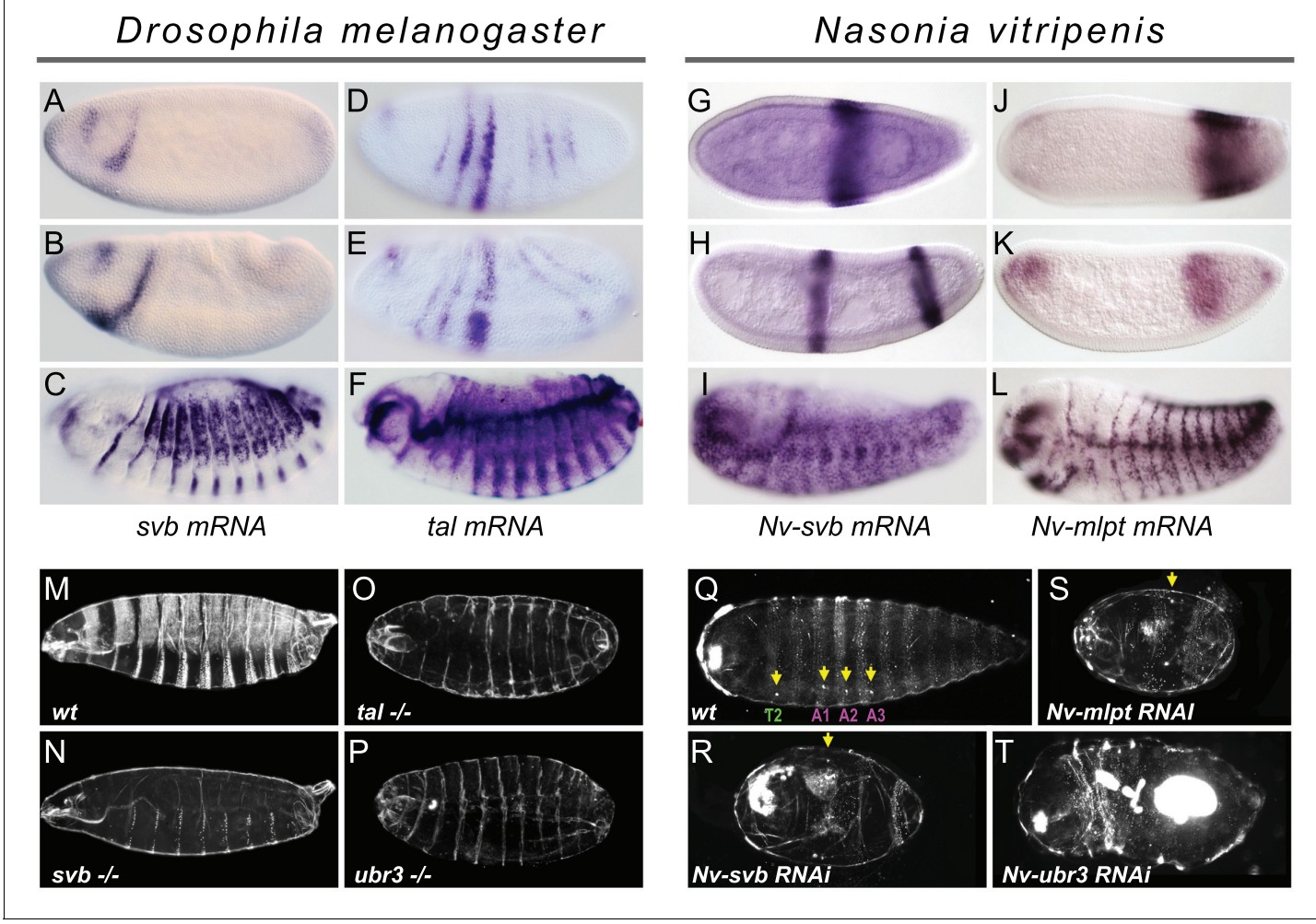

**Figure 5.** Expression and function of *svb*, and *mlpt/tal* in the long germ embryos of *Drosophila melanogaster* and *Nasonia vitripennis*. (A–F) In situ hybridization of *Drosophila* embryo to *svb* (A–C) and *tal/mlpt* (D–F) mRNA. In blastoderm and gastrula embryos, *svb* mRNA is restricted to two stripes in the head (A,B) while *tal* is expressed in seven thin stripes in the presumptive abdomen (D,E). At late embryonic stages, *svb* and *tal* are expressed in epidermal trichome cells (C,F). (G–L) Expression of *Nv-svb* (G–I) and *Nv-mlpt* (J–L) in *Nasonia* embryo. *Nv-svb* is expressed in the mid (G) blastoderm in a single broad stripe, and in the late (H) blastoderm in two stripes. Early *Nv-mlpt* mRNA expression is observed as an anterior cap and a stronger posterior domain (J); anterior expression fades with enrichment of a strong stripe at the posterior as embryogenesis progresses (K). Late *Nasonia* embryos exhibit widespread *Nv-svb* and *Nv-mlpt* expression, with enrichment in a segmental pattern similar to the pattern of trichomes (I, L). (M–P) Cuticles of *Drosophila* young larvae. (M) Wild type larva showing typical pattern of ventral and dorsal trichomes. Embryos lacking maternal and zygotic *tal* (O), *svb* (N), and *ubr3* (P) completely lack embryonic trichomes, and exhibit general cuticle defects. (Q–T) Cuticles of *Nasonia* larvae. (Q) Wild type larva with 4 pairs of spiracles (yellow arrowheads), on thoracic segment T2, and abdominal segments A1, A2 and A3. Cuticles of *Nv-mlpt* (S) and *Nv-svb* (R) RNAi larvae are extremely truncated with loss/fusion of most abdominal segments. Fusion of remaining anterior segments are also detected in *Nv-mlpt* embryos with only one remaining spiracle, *Nv-svb* larva shows fusion of thoracic segments. *Nv-ubr3* RNAi larva exhibit dramatic phenotypes with little or no cuticle. Milder phenotype (T) includes a shortened larva with a thin cuticle decorated with few denticles on the anterior side.
DOI: https://doi.org/10.7554/eLife.39748.022

The following figure supplements are available for figure 5:

**Figure supplement 1.** Maternal and zygotic depletion of *svb*, *tal* or *ubr3* does not affect embryonic segmentation in *Drosophila*.
DOI: https://doi.org/10.7554/eLife.39748.023

**Figure supplement 2.** Expression of *Nv-svb* during embryogenesis.
DOI: https://doi.org/10.7554/eLife.39748.024

**Figure supplement 3.** Detailed expression of *Nv-mlpt* throughout *Nasonia* embryogenesis.
DOI: https://doi.org/10.7554/eLife.39748.025

**Figure supplement 4.** Embryonic expression in *Nasonia* of *dusky-like* and *singed*, two Svb epidermal targets in *Drosophila*.
DOI: https://doi.org/10.7554/eLife.39748.026

**Figure supplement 5.** Phenotypes of increasing strength for *Nv-mlpt*-RNAi, *Nv-svb*-RNAi, *Nv-ubr3*-RNAi in *Nasonia* embryos.

*Figure 5 continued on next page*

*Figure 5 continued*

DOI: https://doi.org/10.7554/eLife.39748.027

*supplement 3*). In both *Nasonia* and *Drosophila*, later expression of *svb* and *mlpt* after germband extension prefigures the pattern of epidermal trichomes (*Figure 5C,F; I,L* and *Figure 5—figure supplements 2* and *3*). Consistent with this observation, we find that several Svb target genes encoding trichome effectors in flies are also expressed with a similar pattern in late *Nasonia* embryos (*Figure 5—figure supplement 4*). Thus, in a wide range of insects, complementary and/or overlapping expression of *svb* and *mlpt* in the embryo correlates with an essential role in embryonic segmentation.

The stereotyped pattern of trichomes (also known as denticles, hairs or microtrichia) is distinctive along the anterior-posterior and dorso-ventral axes, providing a readout for correct segmentation. In flies, although trichomes are severely reduced (hence, 'shaven') in the thin cuticles of mutants for *svb*, *tal*, or *ubr3* (*Figure 5M–P*), all segments are still formed (*Figure 5—figure supplement 1*). In the cuticle of *Nasonia*, the trichome pattern highlights three thoracic segments and 10 abdominal segments; four spiracles (located on thoracic segment T2 and abdominal segments A1- A3) provide landmarks for segment identification (*Pultz et al., 2000*). *Nv-mlpt* RNAi causes posterior truncation and segment fusions, evident as severely shortened larvae, with two remaining trichome belts that likely correspond to thoracic and anterior abdominal segments (*Figure 5Q,S* and *Figure 5—figure supplement 5A–A''*). Similarly, *Nv-svb* RNAi causes severe posterior truncation and loss of most abdominal segments, with only one or two pairs of spiracles left (*Figure 5R* and *Figure 5—figure supplement 5B–B''*). Larvae from *Nv-ubr3* RNAi were almost uniformly too fragile to recover (not shown), likely owing to the observed absence/thinning of cuticle. Mildly affected *Nv-ubr3* RNAi larvae exhibit thin cuticle, devoid of trichomes on the posterior (*Figure 5T* and *Figure 5—figure supplement 5D,D'*).

Altogether, our data support conserved functions for *mlpt*, *svb* and *ubr3* in embryonic segmentation of *Nasonia vitripennis*, a long germ insect, leaving only *Drosophila* from among species tested without such an early patterning function.

## Restoring *svb* expression in the early *Drosophila* embryo disrupts segmentation

Since we find this functional module to be ancestral and deeply conserved in both short and long germ insects, we sought to investigate how the module lost its segmentation role in flies. *Drosophila ubr3* is ubiquitous and *tal* is expressed in pair-rule like stripes, but *svb* expression is absent in the abdomen at early embryonic stages (see *Figure 5*). We therefore hypothesized that the loss of the segmentation function of this module may have involved the loss of *svb* expression during early embryogenesis in the lineage leading to *Drosophila*.

To test this hypothesis, we added back *svb* expression to the early embryo to mimic *svb* early expression that is observed in *Tribolium*, *Oncopeltus*, *Gerris*, and *Nasonia*, using the Gal4/UAS system (*Brand and Perrimon, 1993*). Strikingly, ectopic expression of *svb* in the early embryo (using *nullo*-Gal4) resulted in strong segmentation defects, with no detectable effects on *tal* expression (*Figure 6A–B''*). We also noticed dramatically increased cell death, as also recently reported in activation of segmentation genes (*Crossman et al., 2018*). Similar defects were also observed following maternal ectopic *svb* expression (*Figure 6—figure supplement 1*), albeit with stronger induction of lethality. These results suggest that the loss of *svb* expression prevents segmentation function of the trio during early embryogenesis in flies, and thus indicates that the function of the *tal/svb/ubr3* module in segmentation is contingent upon expression of all three partners.

To further evaluate this conclusion, we tested whether the segmentation defects resulting from Svb ectopic expression involved the function of naturally expressed *tal* and *ubr3* members of the module. To do this, we generated a transgene encoding an N-terminal truncated Svb protein, lacking the N-terminal repression domain, thereby mimicking the shorter Svb activator form (Svb-ACT) that otherwise results from Tal/Ubr3-mediated processing (*Kondo et al., 2010*; *Zanet et al., 2015*). Reciprocally, we engineered a transgenic Svb variant insensitive to processing (*Zanet et al., 2015*), by mutating the 3 Lysine residues that are recognized and ubiquitinated by Ubr3 (Svb-3Kmut). As

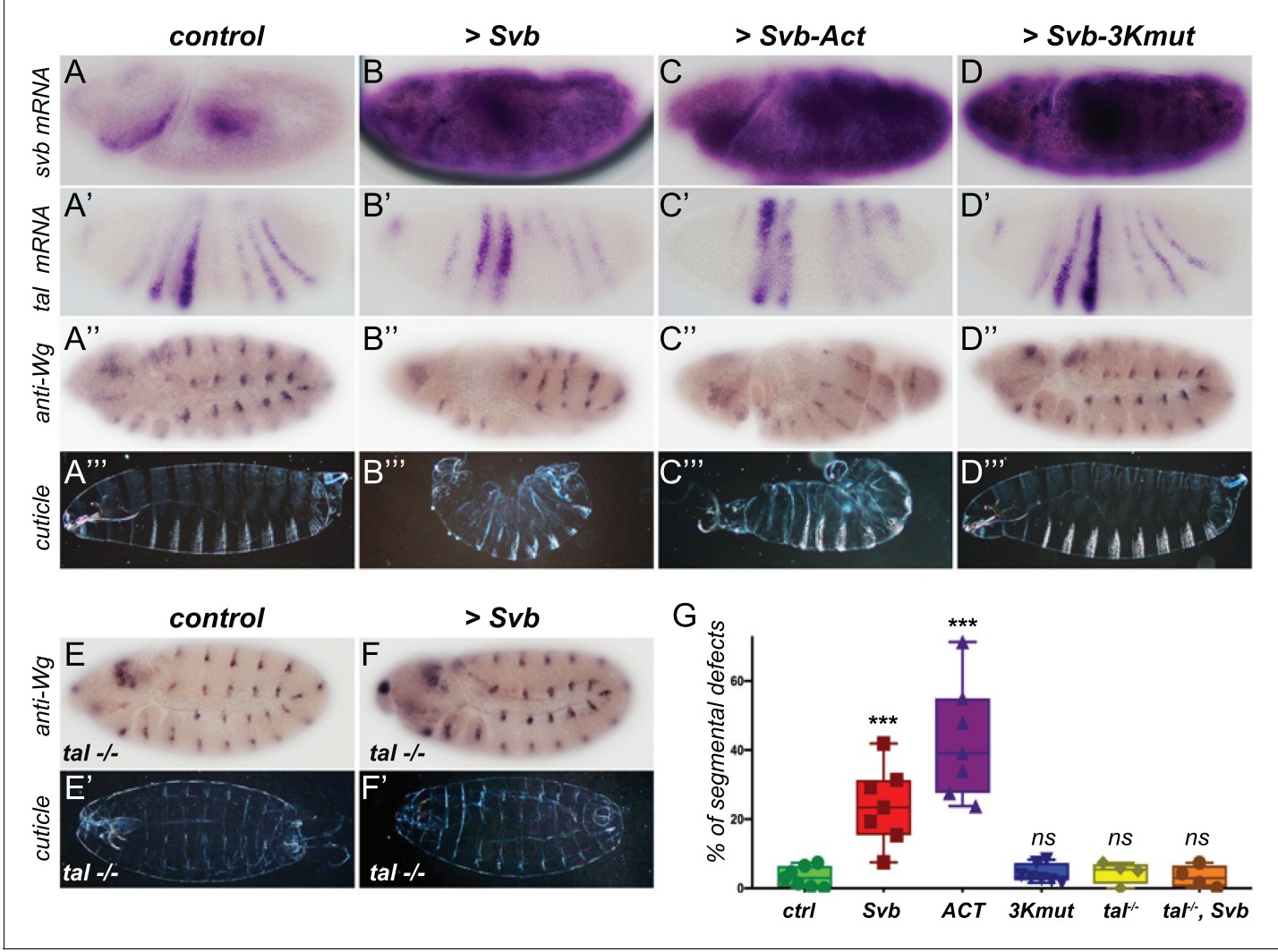

**Figure 6.** Reawakening *svb* expression in the early *Drosophila* embryo affects segmentation. Top panels show in situ hybridization of *svb* (A–D) and *tal* (A'–D') mRNA and anti-Wingless (Wg) immunostaining (A"–D") at gastrulation stage in control conditions (*nullo >GFP*) and following the ectopic expression (driven by *nullo-Gal4*) of wild type Svb (B–B"), Svb-ACT (C–C") and Svb-3Kmut (D–D"), which mimics or prevents Pri/Ubr3-mediated processing of Svb, respectively. (A'''–D''') show cuticle preparations of control (A"), *nullo >Svb* (B"'), *nullo >Svb* ACT (C"') and *nullo >Svb-3Kmut* (D") embryos. (E–F') panels show immunostaining for the Wingless protein and cuticle preparations of control (E–E') and *svb* ectopic expression (*nullo >Svb*) (F–F') in a *tal* null genetic background. *tal* mutant embryos display characteristic trichome loss and cuticle defects. (G) Quantification of segmental defects for each genotype. Data were analyzed by one-way ANOVA. ***, p-value<0,002; *ns*, non-significant. Total numbers of embryos are 177 (*ctrl*), 62 (*Svb*), 621(*Act*), 413 (*3Kmut*), 223 (*tal-/-*) and 138 (*tal-/-, Svb*). Source data for ***Figure 6G*** are found in Source Data File 1.
DOI: https://doi.org/10.7554/eLife.39748.028

The following figure supplements are available for figure 6:

**Figure supplement 1.** Early ectopic expression of *svb* using different drivers triggers segmentation defects.
DOI: https://doi.org/10.7554/eLife.39748.029
**Figure supplement 2.** Effect of modified forms of the Svb protein on epidermal trichome formation.
DOI: https://doi.org/10.7554/eLife.39748.030

expected, the expression of Svb-ACT and Svb-3Kmut in the embryonic epidermis leads to ectopic trichomes and trichome loss, respectively (***Figure 6—figure supplement 2***). When expressed in the early embryo, Svb-Act causes segmentation defects that are reminiscent of those obtained by Svb over-expression, albeit at higher frequency (***Figure 6C–C"***). In contrast, expression of Svb-3Kmut, which is insensitive to Tal/Ubr3, in the early embryo did not cause segmentation defects (***Figure 6D–D"***). These results indicate that the segmentation defects observed upon Svb ectopic expression in

early embryos rely on its processing into the activator form, and that, in this context, ectopic Svb can be regulated by endogenous Tal peptides. To further reinforce this conclusion, we assayed the consequences of Svb ectopic expression in early *Drosophila* embryos lacking *tal* function. Compared to otherwise wild-type embryos, the ectopic expression of Svb in the absence of *tal* failed to cause any detectable defects in segmentation (*Figure 6E–G*), while impaired epidermal differentiation is obvious.

Taken together, these data support the conclusion that the cooperativity of this gene module has remained intact throughout evolution, and that the inactivation of its function in *Drosophila* segmentation involved abrogation of early expression of Svb, an essential component of the module.

## Discussion

Our experiments reveal how a cooperative trio of molecules, initially discovered within a more restricted capacity during terminal epidermal differentiation in *Drosophila*, possesses important ancestral functions in insect embryonic segmentation. These findings represent a significant addition to the anterior-posterior patterning network in insects and provide novel insights into how conserved molecular complexes may contribute to organismal evolution.

Together with the conserved protein structural signature motifs underlying regulatory interactions between Mlpt peptides and Ubr3/Svb proteins, we present evidence for several conserved functions of this module across considerable evolutionary distances. Upon the inactivation of any of the three functional partners, all insects representing both ancestral and derived segmentation modes exhibit strong epidermal defects, evident both in trichome differentiation and in the thinning of the cuticle. The epidermal functions of the module, the most well-described in flies, likely involve the conservation of a similar set of target genes. Several Svb epidermal targets identified in *Drosophila melanogaster* (*Chanut-Delalande et al., 2006*; *Fernandes et al., 2010*; *Menoret et al., 2013*) and sister species (*Chanut-Delalande et al., 2006*) are indeed similarly regulated in *Tribolium* (*Li et al., 2016*). Expression patterns of Svb epidermal target genes in *Nasonia* support a similar conclusion.

A second shared function across all species examined is the importance of Mlpt/Ubr3/Svb for leg specification and patterning, as initially reported in flies (*Galindo et al., 2007*; *Pueyo and Couso, 2008*; *Pueyo and Couso, 2011*). Analysis of more basal insects shows that inactivation of any of the three partners leads to shortened and misdifferentiated legs, often with missing/fused segments, in particular in their distal parts. The conserved outputs of this module highlight transcriptional networks downstream of Svb whose connectivity also appears largely intact over large evolutionary distances (*Spanier et al., 2017*). Together, these data underscore the ancestral conservation of a functional tripartite molecular complex, of its target transcriptional networks and roles in embryonic/post-embryonic development, dating to early in the radiation of arthropods.

Outside of *Drosophila*, we demonstrate function of the module in the formation of posterior segments in all species tested, delineating a key module for insect embryonic segmentation. A strong domain of *svb* expression in the growth zone is observed in all short germ species examined, often adjacent to a strong *mlpt* expression domain. In the long germ wasp embryo, *Nv-mlpt* and *Nv-svb* are also expressed in adjacent/partly-overlapping domains, at the time they function in segmentation. The existence of *mlpt/svb* boundaries may result from mutual exclusivity between *svb* and *mlpt* expression. Such abutting stripes of *mlpt* (*tal* in flies) and *svb* have been described in formation of adult leg joints in flies (*Pueyo and Couso, 2011*). It is worth mentioning that the Mlpt/Svb function in leg joint formation in flies involves Notch-mediated signaling (*Pueyo and Couso, 2011*), a pathway required for coordination of the segmentation clock from basal arthropods (*Chipman and Akam, 2008*; *Eriksson et al., 2013*; *Stollewerk et al., 2003*) to mammals (*Hubaud and Pourquié, 2014*). The Svb/Mlpt expression boundaries at the interface between blastoderm and (oscillation-driven) growth zone in insects thus invites further study, for example to assay whether it might constitute a retracting wavefront (regulated by a speed regulator (*Zhu et al., 2017*)) which is smoothened by Mlpt diffusion and may serve to sharpen and polish expression boundaries of pair-rule genes or other gap genes, a role comparable to that of Notch during somitogenesis (*Dequéant and Pourquié, 2008*).

Beyond insects, Svb (also known as Ovo or OvoL) is conserved in all animals, and predates bilateria (*Kumar et al., 2012*). In addition to the germline and epidermis (*Dai et al., 1998*; *Lee et al., 2014*; *Nair et al., 2006*), recent studies have uncovered a broader role of OvoL/Svb in epithelial

organization and regulation of Epithelial-Mesenchymal Transition (*Bai et al., 2018*; *Kitazawa et al., 2016*; *Lee et al., 2014*; *Nieto et al., 2016*; *Watanabe et al., 2014*). Although the sequential nature of segmentation and posterior segment addition – in both invertebrates and vertebrates – is well known from classical embryology, the cellular mechanisms integrated in the function of the growth zone, that is the contribution of cell division, cell movement, and cytoskeletal reorganization, remain only incompletely understood, including in insects (*Williams and Nagy, 2017*). As in germband elongation of the *Drosophila* embryo (*Collinet et al., 2015*; *Munjal et al., 2015*), which occurs after segmentation in this derived species, the elongation of short germ embryos likely also relies heavily on cytoskeletal rearrangements (*Mao and Lecuit, 2016*). Interestingly, basal insect embryos with reduced *mlpt* or *svb* often appear deficient in convergent extension (*Figure 2* and *Figure 4—figure supplement 4*), suggesting that this module may be involved in the control of cytoskeletal rearrangement during segmentation. The development of suitable tools for live-imaging of cell/cytoskeleton dynamics in a growing number of species (*Auman and Chipman, 2017*; *Benton, 2018*) will offer new ways to investigate the cellular mechanisms of segment addition and to decipher the role of the Mlpt/Ubr3/Svb module therein.

Recent advances in mapping protein-protein interactions at a proteome-wide scale show the unexpected prevalence of ancestral macromolecular complexes, highly conserved across metazoans (*Wan et al., 2015*). Multi-protein complexes appear to evolve more slowly than gene regulatory networks (*Tan et al., 2007*), mirroring deep conservation of protein-protein interaction domains across orthologues. How might ancient protein complexes that are evolutionarily stable throughout animals nevertheless undergo phenotypic diversification and incorporate novelty? Our data show that Ubr3 is required for the activity of the complex, but its function is clearly permissive, as seen by ubiquitous expression across species. In contrast, the dynamic patterns of *mlpt/ta*l and *svb* highlight the key aspect of the control of their expression. Evolutionary changes in enhancers and associated trans-acting factors of these two instructive members of the module likely underlie evolution of their function in segmentation. Svb enhancers are well-documented for their modifications across *Tephritidae* and *Drosophilidae*, which are causal for the evolution of trichome pattern (*Frankel et al., 2011*; *Frankel et al., 2012*; *Khila et al., 2003*; *McGregor et al., 2007*; *Preger-Ben Noon et al., 2016*; *Sucena et al., 2003*). A similar change in promoter control of Svb expression may be sufficient to bring segment patterning potency on- or off-line in the insect embryo. The phylogenetic distribution within insects of short/long germ modes of development implies that evolution has repeatedly sampled these modes (*Misof et al., 2014*). Recent data support a model in which segmentation mechanisms in short and long germ insects are more similar than initially thought (*Benton, 2018*; *Clark, 2017*), and mostly differ in the specifics of their timing (*Zhu et al., 2017*). Our data suggest one mechanism by which delayed posterior segment formation may be switched on/off via Svb/Mlpt/Ubr3.

Together, our data suggest how integration of a post-translational mechanism involving a micropeptide like Mlpt can be used in combination with transcriptional control to regulate Svb, both in protein activity and expression timing, to broadly regulate phenotypic plasticity during embryogenesis. This suggests future research directions integrating insights from evolution of transcriptional regulation and micropeptide discovery into the functional study of multi-protein complexes, to facilitate the elucidation of mechanisms of and constraints upon organismal evolution.

## Materials and methods

**Key resources table**

| Reagent type (species) or resource | Designation | Source or reference | Identifiers | Additional information |
|---|---|---|---|---|
| Gene (*Drosophila melanogaster*) | ovo/svb | NA | FLYB: FBgn0003028 | |
| Gene (*Drosophila melanogaster*) | tal | NA | FLYB: FBgn0087003 | |

*Continued on next page*

*Continued*

| Reagent type (species) or resource | Designation | Source or reference | Identifiers | Additional information |
|---|---|---|---|---|
| Gene (*Drosophila melanogaster*) | Ubr3 | NA | FLYB: FBgn0260970 | |
| Gene (*Tribolium castaneum*) | Tc-svb | this paper | Genbank: MG913606 | |
| Gene (*Tribolium castaneum*) | mlpt | NA | GenBank: AM269505.1 | |
| Gene (*Tribolium castaneum*) | Tc-Ubr3 | NA | NCBI Ref S eq: XM_964327 | beetlebase: T C005949 |
| Gene (*Oncopeltus fasciatus*) | Of-svb | this paper | GenBank: MH181832 | |
| Gene (*Oncopeltus fasciatus*) | Of-mlpt | this paper | GenBank: MH181830 | |
| Gene (*Oncopeltus fasciatus*) | Of-Ubr3 | this paper | GenBank: MH181827 | |
| Gene (*Gerris buenoi*) | Gb-svb | this paper | GenBank: MH011417 | |
| Gene (*Gerris buenoi*) | Gb-mlpt | this paper | GenBank: MH699965 | |
| Gene (*Gerris buenoi*) | Gb-Ubr3 | this paper | GenBank: MH011418 | |
| Gene (*Nasonia vitripennis*) | Nv-svb | this paper | GenBank: MH181831 | |
| Gene (*Nasonia vitripennis*) | Nv-mlpt | this paper | GenBank: MH181829 | |
| Gene (*Nasonia vitripennis*) | Nv-Ubr3 | this paper | GenBank: MH181828 | |
| Strain, strain background (*Nasonia vitripennis*) | AsymCx | PMID: 20075255 | | |
| Genetic reagent (*D. melanogaster*) | FM7C, Kr > GFP | Bloomington Drosophila Stock Center | BDSC: 5193; FLYB: FBst0005193; RRID:BDSC_5193 | FlyBase symbol: Df(1) JA27/FM7c, P{w[+mC] =GAL4 Kr.C}DC1, P{w[+mC] =UAS GFP.S65T} DC5, sn[+] |
| Genetic reagent (*D. melanogaster*) | TM6B, ubi-GFP | Bloomington Drosophila Stock Center | BDSC: 4887; FLYB: FBst000 4887; RRID: BDSC_4887 | FlyBase symbol: w[1118]; Df(3L)Ly, sens[Ly-1]/TM6B, P{w[+mW.hs]=Ubi GFP.S65T}PAD2, Tb[1] |

*Continued on next page*

*Continued*

| Reagent type (species) or resource | Designation | Source or reference | Identifiers | Additional information |
|---|---|---|---|---|
| Genetic reagent (D. melanogaster) | TM3, twist-GAL4 > GFP | Bloomington Drosophila Stock Center | BDSC: 6663; FLYB: FBst0006663; RRID:BDSC_6663 | FlyBase symbol: w[1118]; Dr[Mio]/TM3, P{w[+mC]=GAL4 twi.G}2.3, P{UAS-2x EGFP}AH2.3, Sb[1] Ser[1] |
| Genetic reagent (D. melanogaster) | nullo-GAL4 | Bloomington Drosophila Stock Center | BDSC:26875; FLYB:FBtp0018484; RRID:BDSC_26875 | FlyBase symbol: P{nullo-GAL4.G}5.20 |
| Genetic reagent (D. melanogaster) | nos-GAL4 | Bloomington Drosophila Stock Center | BDSC:4937; FLYB:FBtp0001325; RRID:BDSC_4937 | FlyBase symbol: P{GAL4::VP16-nos.UTR} CG6325MVD1 |
| Genetic reagent (D. melanogaster) | ptc-GAL4 | Bloomington Drosophila Stock Center | BDSC:2017; FLYB:FBti0002124; RRID:BDSC_2017 | FlyBase symbol: P{GawB}ptc559.1 |
| Genetic reagent (D. melanogaster) | pri[1] | PMID:17486114 | FLYB: FBal0198099 | Flybase symbol: talS18 |
| Genetic reagent (D. melanogaster) | tal[S18.1] | PMID:17486114 | FLYB:FBal0241050 | Flybase symbol: talpri-1 |
| Genetic reagent (D. melanogaster) | pri[4] | gift from Y. Kageyama | | |
| Genetic reagent (D. melanogaster) | pri[5] | gift from Y. Kageyama | | |
| Genetic reagent (D. melanogaster) | svb[R9] | PIID: 12915226 | FLYB:FBal0151651 | Flybase symbol: ovo[svb-R9] |
| Genetic reagent (D. melanogaster) | ovo[D1] | PMID: 17246182 | BDSC: 23880; FLYB: FBst0023880; RRID:BDSC_23880 | Flybase symbol: ovo[D1] |
| Genetic reagent (D. melanogaster) | svb[PL107] | PMID: 11744370 | DGGR:106675; FLYB: FBst0305341; RRID:DGGR_106675 | Flybase symbol: ovo[PL107] |
| Genetic reagent (D. melanogaster) | Ubr3B | PMID: 26383956 | FLYB:FBal0013375 | Flybase symbol: Ubr3[B] |
| Genetic reagent (D. melanogaster) | UAS-GFP | Bloomington Drosophila Stock Center | FLYB:FBal0129171 | FlyBase symbol: w[*]; P{w[+mC]=UAS GFP .S65T} Myo31 DF[T2] |
| Genetic reagent (D. melanogaster) | UAS-svb::GFP | PMID: 20647469 | FLYB: FBal0319860 | FlyBase symbol: ovoUAS.svb.GFP |
| Genetic reagent (D. melanogaster) | UAS-pri | PMID: 17486114 | BDSC: 1521; FLYB:FBti0003040; RRID:BDSC_1521 | FlyBase symbol: talUAS.cKa |
| Genetic reagent (D. melanogaster) | UAS-svbACT::GFP | this paper | FLYB:FBal0248431 | |
| Genetic reagent (D. melanogaster) | UAS-svb-3Kmut::GFP | this paper | FLYB:FBal0241056 | |

*Continued on next page*

*Continued*

| Reagent type (species) or resource | Designation | Source or reference | Identifiers | Additional information |
|---|---|---|---|---|
| Antibody | anti-Wingless | Developmental Studies Hybridoma Bank | | (1:100) |
| Antibody | anti-Ubx-AbdA | Developmental Studies Hybridoma Bank | | (1:5) |
| Antibody | anti-Dll abbit polyclonal | | DSHB Cat# 4d4; RRID:AB_528512 | (1:200) r |
| Antibody | anti-Dig AP Fap (polyclonal sheep) | Roche | DSHB Cat# UBX/ABD-A FP6.87; RRID: AB_10660834 | (1:2000) |
| Antibody | anti-mouse-HRP (rabbit polyclonal) | Promega | | (1:1000) |
| Antibody | anti-rabbit-HRP (donkey polyclonal) | Jackson Immuno Research | Roche Cat# 11093274910; RRID:AB_514497 | (1:500) |
| Antibody | anti-mouse biotinylated (goat polyclonal) | Vector Laboratories | Promega Cat# W4011; RRID:AB_430833 | (1:500) |
| Recombinant DNA reagent | pUASp-Svb::GFP (plasmid) | PMID:17486114 | Jackson ImmunoResearch Labs Cat# 711-035-152; RRID:AB_10015282 | |
| Recombinant DNA reagent | pUASp-SvbAct ::GFP (plasmid) | this paper | Vector Laboratories Cat# BA-9200; RRID:AB_2336171 | Progenitors: PCR, pUASp-Svb::GFP |
| Recombinant DNA reagent | pUASp-Svb-3 Kmut::GFP (plasmid) | this paper | | Progenitors: pAc-SvbK7; pUASp-Svb::GFP |
| Recombinant DNA reagent | pCR-Topo (plasmid) | Qiagen | | |
| Recombinant DNA reagent | pBluescript (plasmid) | Stratagene | | |
| Recombinant DNA reagent | pGEM-Teasy (plasmid) | Promega | Quiagen Cat#: 231122 | |
| Recombinant DNA reagent | pBac (3xP3-EGFPafm) (plasmid) | gift from E. Wimmer | Stratagene Cat#: 212205 | Flybase symbol: PBac {3xP3-EGFPafm} |
| Recombinant DNA reagent | pBME (TcU6b-Bsal) (plasmid) | gift from A. Giles | Promega Cat#: A1360 | Original gRNA expression vector with Bsa1 sites |
| Recombinant DNA reagent | pSLfa(Hsp-p-nls-Cas9-3'UTR)fa (plasmid) | gift from A. Giles | FLYB: FBtp0014061 | Cas9 expression vector |
| Recombinant DNA reagent | Tc-U6b-sim ZS1 (plasmid) | Rode and Klingler, unpublished | | *sim* gRNA expression vector |
| Sequence-based reagent | see *Supplementary file 1B* for a complete list of oligonucleotides used in this paper | | | |

*Continued on next page*

*Continued*

| Reagent type (species) or resource | Designation | Source or reference | Identifiers | Additional information |
|---|---|---|---|---|
| Commercial assay or kit | DIG RNA Labeling kit | Roche | | |
| Commercial assay or kit | NBT-BCIP solution | Roche | | |
| Commercial assay or kit | In-Fusion HD Cloning Kit | Clontech | Roche Cat#: 11 277 073 910 | |
| Commercial assay or kit | MEGAscript RNA kit | ThermoFischer | Sigma Cat#: 72091 | |
| Chemical compound, drug | Blocking reagent | Roche | Takara Cat#: 21416 | |
| Chemical compound, drug | 3,3'-Diaminobenzidine tetrahydro chloride hydrate | Sigma | ThermoFischer Cat#: AM1626 | |
| Software, algorithm | Next-RNAi | http://www.nextrnai.org | Roche Cat#: 11 096 176 001 | |
| Software, algorithm | Primer3 | https://primer3plus.com | Sigma Cat#:32750 | |
| Software, algorithm | MacVector | https://macvector.com | | |
| Software, algorithm | Prism 8 | https://www.graphpad.com/ | Primer3Plus; RRID:SCR_003081 | |
| Software, algorithm | Photoshop CC 2019 | https://www.adobe.com/ | MacVector; RRID:SCR_015700 | |
| Software, algorithm | Illustrator CC 2019 | https://www.adobe.com/ | GraphPad Prism; RRID:SCR_002798 | |
| Software, algorithm | Acrobat Pro DC | https://www.adobe.com/ | Adobe Photoshop; RRID:SCR_014199 | |
| Software, algorithm | Axiovision 4.6.3.SP1 | Zeiss | Adobe Illustrator; RRID:SCR_010279 | |

## Tribolium castaneum

Insects were reared at ambient temperature of 25°C. Embryos were collected and whole-mount in situ hybridization performed as previously described (*Patel et al., 1989*; *Schinko et al., 2009*; *Tautz and Pfeifle, 1989*). Digoxigenin- labelled RNA probes were detected using alkaline phosphatase-conjugated anti-DIG antibodies (1:2000; Roche) and NBT/BCIP substrates (Roche), as per manufacturer's instructions. Sequence of all oligonucleotides used in this study (for the five insect species) is given in *Supplementary file 1B*.

Double-stranded RNA synthesis and parental injection were performed as described previously (*Bucher and Klingler, 2004*; *Bucher et al., 2002*). dsRNAs were injected into female pupae or virgin adult females at a concentration of 1–3 µg/µl. RNAi phenotypes were confirmed by using non-overlapping dsRNA fragments for each gene. First instar larval cuticles were cleared in Hoyer's medium/lactic acid (1:1) overnight at 60°C. Cuticle auto-fluorescence was detected on a Zeiss Axiophot microscope. Z stacks and projections were created with a Zeiss ApoTome microscope using the Axiovision 4.6.3.SP1 Software. Color images were taken by (ProgResC14) using the ProgResC141.7.3 software and maximum projection images were created from z stacks using the Analysis D software (Olympus).

For *Tribolium svb*, all primer pairs shown were used to generate template for dsRNA synthesis. Amplicons generated by the last four pairs were also used for antisense RNA probe synthesis. dsRNA fragments corresponding to different regions of the *svb* transcript were used for gene knockdown by RNAi. All dsRNA fragments resulted in similar knockdown phenotypes with high penetrance. Primers were designed based on the Next-RNAi software, Primer3 or MacVector. The nucleotides shown in red indicate tags of parts of T7 (3' primer) and SP6 (5' primer) promoter

sequences attached to gene-specific sequences in the manner described by *Schmitt-Engel et al. (2015)*. The products were used for a second PCR using T7 and SP6-T7 primers for generating a double stranded template for in vitro transcription by T7 polymerase. For in situ RNA probes, the second PCR was done using the complete T7 and SP6 promoter sequences and subsequently in vitro transcription was performed to generate a Digoxigenin-labelled antisense RNA probe with the appropriate polymerase. Amplicons that were cloned into pBluescript vector were amplified with T7 and T3-T7 primers for subsequent dsRNA synthesis or T7 and T3 primers for subsequent antisense RNA probe synthesis using either T3 or T7 RNA polymerase. The primer design was based on the RNAseq data (Tcas au5 prediction) for *Tc-svb* available on iBeetle-Base. For *mlpt* dsRNA and probe synthesis, a full-length *mlpt* cDNA cloned into pBluescript was obtained from Dr. Michael Schoppmeier. For *Tc-ubr3*, all primer pairs shown were used for dsRNA synthesis. All dsRNA fragments resulted in similarly strong knockdown phenotypes with very high penetrance. The fragments generated with the primers containing iBeetle numbers were also used as probes.

To generate a *Tc-svb* mutant using CRISPR/Cas9, gRNAs were directed to the putative transactivation domain in exon 2 of *Tc-svb*. The sequence of primers used is given in *Supplementary file 1C*, with the G (required by the U6 promoter for transcription initiation) marked in green, the PAM sequence in blue, and the sequences in orange representing the complementary overhangs generated by Bsa1 digestion. A fourth gRNA was directed to the *Tribolium single-minded* gene (*Tc-sim*, Rode and Klingler, unpublished). Embryonic injection mix consisted of 125 ng each of the four gRNA expression vectors, 500 ng of the donor eGFP vector containing the sim target sequence, and 500 ng of the Cas9 expression vector. Non-homologous end joining (NHEJ) method was employed for directed knock-in of an eGFP containing donor marker plasmid (*Supplementary file 1D*) into the exon2 of the endogenous *Tc-svb* gene. The *sim* gRNA was used to target the *sim* sequence in the marker plasmid leading to its Cas9-induced linearization. This was followed by insertion of the linearized plasmid into one or more target sites in the *Tc-svb* genome. A successful knock-in of the marker plasmid was obtained only at gRNA target site 3. This insertion site was present in all *Tc-svb* transcripts and was also downstream from a putative second start codon, thus increasing the chances of obtaining a *Tc-svb* null phenotype.

## Oncopeltus fasciatus

Wild-type *Oncopeltus* embryos were collected on cotton from mated females, and aged, as needed, in a 25°C incubator. Embryos were first boiled for 1 to 3 min in a microfuge tube in water, followed by a 1 min incubation on ice, before further processing. Embryos were fixed in 12% heptane-saturated formaldehyde/1X PBS for 20 min with shaking. The heptane was replaced by methanol, and the embryos either stored under methanol at −20°C or processed immediately. Embryos were then rehydrated to 1X PBT through a methanol/PBT series, and dechorionated, before further fixation for 90–120 min in 4% formaldehyde/1X PBT. Embryos were then transferred to and stored in 100% methanol.

In situ hybridizations were carried out (as described for *Nasonia* in *Rosenberg et al., 2014*) on embryos peeled and stored under 100% methanol, and rehydrated through an methanol/1x PBS, 0.1%Tween (1xPBT) series. Briefly, rehydrated embryos were washed several times in 1x PBT before a 5 min post-fix in 5% formaldehyde/1X PBT, followed by 3 five minutes washes in 1X PBT. Embryos were briefly treated with Proteinase K (4 µg/ml final concentration) in 1X PBT for 5 min, followed by 3 five minute washes in 1X PBT, and an additional 5 min post-fix in 5% formaldehyde/1X PBT. Following 3 x three minute washes in 1X PBT, embryos were incubated in hybridization buffer for 5 min at room temperature, followed by incubation in fresh hybridization buffer for a 1 hr pre-hybridization step at 65°C. RNA probes were prepared and added to a fresh portion of hybridization buffer and incubated at 85°C for 5 min, then one minute on ice, before replacing pre-hybridization with hybridization buffer containing denatured RNA probe. Tubes were incubated overnight at 65°C. After washes in formamide wash buffer, embryos were washed in several changes of 1X MABT buffer, before incubation in 1X MABT +2% Blocking Reagent (BBR; Roche) for 1 hr, and then 1X MABT/2% BBR/20% sheep serum for an additional hour, before addition of fresh 1X MABT/2%BBR/20% sheep serum containing anti-DIG AP Fab fragments (1:2000; Roche) for overnight incubation at 4°C. In the morning, extensive 1X MABT washes were carried out before equilibration of embryos with AP staining buffer and then staining with AP staining buffer containing NBT/BCIP (Roche; as per manufacturer's instructions). After staining, three 1X PBT washes were carried out before a final post-fixation

step (5% formaldehyde/1xPBT), and then one PBT wash before sinking in 50% glycerol/1xPBS, and then 70% glycerol/1xPBS, which was also used for mounting before imaging.

dsRNA templates were amplified from target gene fragments which had been cloned into either pCR-Topo (Qiagen) or pGEM (Invitrogen), using T7 promoter-containing oligos, as described previously (*Lynch and Desplan, 2006*). Purified PCR product was used for dsRNA transcription using Megascript RNAi (Ambion) according to manufacturer's instructions. dsRNA was injected into newly eclosed virgin female milkweed bugs, at a concentration of 1–3 µg/µl. After injection, females were mated to uninjected males, and embryos were collected for the duration of egg laying. Embryos for phenotypic evaluation were incubated at 28°C for 8 days, and unhatched embryos were dissected from their membranes and imaged for phenotypes.

## Gerris buenoi

Wild type *Gerris buenoi* were collected from a pond in Toronto, Ontario, Canada and established in the lab. Stocks were maintained in aquaria at 25°C with a 14 hr light/10 hr dark cycle, and fed with fresh crickets. Styrofoam float pads were provided to females as substrate for egg laying. Embryos were collected and incubated at 20–25°C until desired developmental time points, at which time they were dissected in 1x PBS with 0.05% tween-20 ('PTW'). Once dissected, embryos were fixed in 4% paraformaldehyde and stored under 100% Methanol at −20°C until use.

In situ hybridizations in *Gerris* were performed as previously described (*Refki et al., 2014*). Briefly, embryos were rehydrated to 1X PBT, through a MeOH/PTW series, and then washed 3 times in PTW to eliminate residual methanol. Embryos were then permeabilized in PBT 0.3% and PBT 1% (1X PBS; 0.3% or 1% Triton X100) for 1 hr. Following these washes, embryos were rinsed once for 10 min in a 1:1 mixture of PBT 1% and hybridization buffer (50% Formamide; 5% dextran sulfate; 100 mg/ml yeast tRNA; 10X salts). The 10X salt mix contains 3 M NaCl; 100 mM Trizma Base; 60 mM NaH2PO4; 50 mM Na2HPO4; 5 mM Ficoll; 50 mM PVP; and 50 mM EDTA. RNA probes corresponding to each gene were transcribed from cDNA templates cloned into pGEM-T (Promega), and then transcribed in vitro using either T7 or Sp6 RNA polymerase (Roche) and labelled with Digoxigenin-RNA labelling mix (Roche). Pre-incubation of embryos was carried out in hybridization buffer for 1 hr at 60°C before adding Digoxigenin-labelled RNA probes overnight at 60°C. The next day, embryos were washed in decreasing concentrations of hybridization buffer diluted with PBT 0.3% (with 3:1, 1:1, 1:3) and then rinsed three times 5 min each in PBT 0.3% and then once for 20 min in blocking solution (1X PBS; 1% Triton X100; 1% BSA) at room temperature before adding alkaline phosphatase conjugated anti-DIG antibody (Roche). Embryos were incubated with primary antibody for 2 hr at room temperature. Following primary antibody, embryos were washed for 5 min in PBT 0.3% and then 5 min in PTW 0.05%. Color enzymatic reaction was carried out using NBT/BCIP substrate (Roche) in alkaline phosphatase buffer (0.1M Tris/HCl pH 9.5; 0.05M MgCl2; 0.1M NaCl; 0.1% Tween-20), according to manufacturer's instructions. Upon completion, the reaction was stopped with several washes of PBT 0.3% and PTW 0.1% (1xPBS; 0.1% Tween-20). Stained embryos were stored in 50% Glycerol/1x PBS at 4°C or −20°C until mounting on slides in 80% glycerol for imaging.

For immunostaining, embryos were cleaned with four times diluted bleach solution and washed in PTW 0.05%. After dissection, embryos were fixed for 20 min in 4% Formaldehyde/1X PTW 0.05%. Embryos were then permeabilized with PBT 0.3% for 30 min and incubated in antibody blocking solution (1X PBS; 0.1% Triton X100; 0.1% BSA; 10% NGS) at room temperature for 1 hr. Embryos were transferred to blocking solution containing primary antibody: mouse anti-Ubx-AbdA, Hybridoma Bank (1:5); rabbit anti-Dll (1:200) and incubated overnight at 4°C. The next day embryos were washed in PTW 0.05% (two quick rinses, then two washes of 10 min each) and incubated for 30 min in blocking solution at room temperature with shaking, before adding the secondary antibody (Rabbit anti-mouse-HRP [1:1000] from Promega or donkey anti-Rabbit-HRP [1:500] from Jackson Immuno research) diluted in PTW. All secondary antibodies were incubated with embryos for 2 hr at room temperature with shaking. Following antibody incubation, embryos were rinsed in PBT 0.3% and PTW 0.05% three times each for 10 min at room temperature. Before enzymatic developing with DAB with color enhancer (DiAminoBenzidine tetrahydro-chloride from Sigma), embryos were briefly incubated with DAB solution for 5 min at room temperature. Upon completion, staining was stopped by washing the embryos briefly in PBT 0.3%, followed by 5 times, five minute washes in PBT 0.3%. Five more washes of 5 min in PTW 0.1% followed. Embryos were transferred to 30% glycerol/1X PBS

for 5 min, and then 50% Glycerol/1X PBS for 5 min, before sinking in 80% glycerol/1X PBS at 4°C until mounting in 80% glycerol under coverslips for imaging.

dsRNA template preparation and injections were carried out as described in *Refki et al. (2014)* and *Santos et al. (2015)*. Briefly target gene fragments were first cloned into pGEM-T vector then amplified using forward and reverse primers tagged with T7 promoter. The resulting PCR product was used for dsRNA transcription using Megascript RNAi (Ambion) according to manufacturer's instructions. dsRNA was injected into adult females at a concentration of 1–3 µg/µl. After injection, females were kept in water containers to lay eggs. Embryos were collected for phenotypic evaluation and imaged for phenotypes.

## Nasonia vitripennis

Wild type *Nasonia* embryos were collected from virgin AsymCx (*Werren et al., 2010*) females host fed on *Sarcophaga bullata* pupae (Carolina Biological), aged as needed at 25°C, and fixed for 28 min in 4% heptane-saturated formaldehyde/1X Phosphate Buffered Saline (PBS), with vigorous shaking. Embryos were hand-peeled under 1X PBT using 1 ml insulin needles (Becton-Dickinson), and were transferred to 100% methanol for storage, or further processed. For staining, embryos were then rehydrated to 1X PBS with 0.1% Tween (PBT) through a methanol/PBT series.

In situ hybridizations were carried out as described previously (*Pultz et al., 2005*; *Rosenberg et al., 2014*). Briefly, fixed embryos that had been stored under methanol were gradually brought up to 1X PBT in a PBT/MeOH series, and washed three times in 1x PBS + 0.1% tween-20 (PBT) before a 30 min post-fixation in 5% formaldehyde/1XPBT. The embryos were then washed three times in 1X PBT, and digested in Proteinase K (final concentration of 4 µg/ml) for five minutes, before three PBT washes. Embryos were blocked for 1 hr in hybridization buffer before probe preparation (85°C, 5 min; ice 1 min) and addition for overnight incubation at 65°C. The next day, embryos were washed in formamide wash buffer three times, and then 1X MABT buffer three times, before blocking in 2% Blocking Reagent (BBR; Roche) in 1X MABT for 1 hr, then in 10% horse serum/2% BBR/1XMABT for 2 hr. Embryos were incubated overnight at 4°C with primary antibody (anti-DIG-AP Fab fragments; Roche, 1:2000). The third day, embryos were washed in 1X MABT for ten x 20 min washes before equilibrating the embryos in AP staining buffer and developing in AP buffer with NBT/BCIP solution (Roche). After staining, embryos were washed in 1x PBT three times for five minutes each before a 25 min post-fix step in 5% formaldehyde/1XPBT. Embryos were then washed several times with 1X PBT, and allowed to sink in 50% glycerol/1XPBS and then 70% glycerol/1XPBS, which was subsequently used for mounting.

dsRNA template was amplified from target gene fragments that had been previously cloned into pCR-Topo (Qiagen) or directly from embryo cDNA, using standard T7 promoter-containing oligos, as described previously (*Lynch and Desplan, 2006*). Purified PCR product was used for dsRNA transcription using Megascript RNAi (Ambion) according to manufacturer's instructions, and purified product diluted to 1–3 µg/µl for injection. pRNAi for *Nv-mlpt* and *Nv-svb* resulted in sterility. Therefore, embryos laid by unmated host-fed virgin *Nasonia* females were microinjected with dsRNA using a Femto-Jet micro-injector (Eppendorf), and transferred to a slide to develop in a humid chamber at 28°C for 36 hr. Unhatched larvae were dissected from extraembryonic membranes and cleared in freshly prepared Lacto:Hoyer's medium overnight at 65°C, and imaged for cuticle organization the following day.

## Drosophila melanogaster

The following *Drosophila* lines were used in this study: *w, pri¹/TM6B-Ubi-GFP* (*Kondo et al., 2007*), *svbR9/FM7-Kr::GFP* (*Delon et al., 2003*), *nullo-Gal4* (from the Gehring lab), *mat-Gal4, nos-Gal* (gift from N. Dostatni). *talpri4, FRT82B/TM6B* and *talpri5, FRT82B/TM6B*, bearing a deletion of the *tal/pri* gene, were kindly provided by Y. Kageyama (Kobe, Japan). UAS constructs used in this study are as follows: *UAS-svb::GFP* (*Kondo et al., 2010*), *UAS-GFP* (Bloomington stock center), *UAS-pri* (*Kondo et al., 2007*), and UAS-svb-ACT::GFP and UAS-svb3Kmut::GFP (this study).

*Ubr3* mutant embryos deprived of maternal and zygotic contribution were generated using the *Ubr3B* allele according to (*Zanet et al., 2015*). Embryos lacking both maternal and zygotic contribution of *pri/tal* were collected from adult females of the following genotype *hsFlp; talS18.1, FRT82B/OvoD1, FRT82B* that received one pulse of heat shock at 37°C for 40 min, during L1-L2 larval stage,

and crossed to males *tal^pri4*, *FRT82B/TM6B-Twist-Gal4,UAS-GFP*. Mutant embryos, identified by the lack of GFP, were sorted and further analyzed. *svb* mutant embryos lacking maternal contribution and/or zygotic contribution were generated by crossing *svb^PL107*, *FRT19A/ovoD1*, *FRT19A*, *hsFlp* adult females that were heat-shocked one hour at 37°C at L1-L2 larval stage to wild type adult males.

To test the effect of *svb* ectopic expression in early embryos lacking *mlpt/pri/tal* function (*tal^pri5*/*tal^S18* trans-heterozygote condition), we generated the following recombinants lines: *tal^pri5*, *nullo-Gal4/TM3*, *Twist-Gal4*, *UAS-GFP*; *tal^S18*, *nullo-Gal4/TM3*, *Twist-Gal4*, *UAS-GFP*; *tal^pri5*, *UAS-svb/TM3*, *Twist-Gal4*, *UAS-GFP*; *tal^S18*, *UAS svb/TM3*, *Twist-Gal4*, *UAS-GFP*. Homozygous *pri/tal* mutant embryos were identified by the lack of balancer chromosome (marked with GFP). Sibling controls and mutant embryos were in all cases processed in the same batch; a typical collection includes >400 embryos in total. Expression of *UAS-svb* constructs using Gal4 drivers were conducted at 29°C.

## DNA constructs and transgenics

To generate the transformation vector pUASp-SvbAct::GFP, a fragment without the exon1S and the 5' of the exon2A to the proteolytic cleavage site was amplified by PCR from pUASp-Svb::GFP (*Kondo et al., 2010*) and cloned into the pUASp-Svb::GFP, linearized with SpeI and EcoRI, using the In-Fusion HD Cloning kit (Clontech). To obtain the pUASp-Svb-3Kmut-GFP, the EcoRI fragment with the 3 K mutated from pAc-SvbK7 (*Zanet et al., 2015*) was cloned into the pUASp-Svb::GFP, linearized with EcoRI. All constructs have been verified by sequencing. Transformation vectors have been used to establish PhiC31-mediated transgenic lines, using standard procedures (*Bischof et al., 2007*).

For embryo staining, staging of mutant embryos, subjected to in situ hybridization or immunohistochemistry, was determined according to the age of embryo collections. Staining was performed as previously described (*Chanut-Delalande et al., 2014*) using: anti-Wg (1/100 mouse monoclonal antiserum, 4D4 Developmental Studies Hybridoma Bank, Iowa City, IA), biotinylated goat anti-mouse (1/500, Vector Laboratories). DIG-labeled RNA antisense probes were synthesized in vitro from cDNA clones and processed for in situ hybridization.

## Data and materials availability

Sequences presented in this paper can be found in Genbank, with accession numbers as follows: *Tc-svb* MG913606, *Nv-mlpt* MH181829, *Nv-Svb* MH181831, *Nv-Ubr3* MH181828, *Of-mlpt* MH181830, *Of-svb* MH181832, *Of-Ubr3* MH181827, *Gb-svb* MH011417, *Gb-mlpt* MH699965, *Gb-Ubr3* MH011418.

## Acknowledgments

We would like to thank Ariel Chipman, Claude Desplan, and Igor Ulitsky for warm support during the course of this work, and for discussions and critical reading of the manuscript, Julien Favier, Amélie Destenabes and Cleopatra Tsanis for embryo injection and help with transgenics.

## Additional information

### Funding

| Funder | Grant reference number | Author |
| --- | --- | --- |
| European Research Council | 616346 | Abderrahman Khila |
| H2020 Marie Skłodowska-Curie Actions | 654719 | Miriam I Rosenberg |
| Fulbright Association | | Miriam I Rosenberg |
| Deutsche Forschungsgemeinschaft | KL656-5 | Martin Klingler |
| Friedrich-Alexander-Universität Erlangen-Nürnberg | | Martin Klingler |

| Agence Nationale de la Re-cherche | ChronoNet | François Payre |
| Fondation pour la Recherche Médicale | DEQ20170336739 | François Payre |

The funders had no role in study design, data collection and interpretation, or the decision to submit the work for publication.

### Author contributions

Suparna Ray, Conceptualization, Data curation, Formal analysis, Supervision, Validation, Investigation, Visualization, Methodology, Writing—original draft, Writing—review and editing; Miriam I Rosenberg, Conceptualization, Data curation, Formal analysis, Supervision, Funding acquisition, Validation, Investigation, Visualization, Writing—original draft, Project administration, Writing—review and editing; Hélène Chanut-Delalande, Formal analysis, Validation, Investigation, Visualization, Writing—review and editing; Amélie Decaras, Conceptualization, Formal analysis, Investigation, Visualization, Writing—review and editing; Barbara Schwertner, Resources, Methodology; William Toubiana, Data curation, Formal analysis, Investigation, Writing—review and editing; Tzach Auman, Formal analysis, Investigation, Visualization, Writing—review and editing; Irene Schnellhammer, Data curation, Formal analysis, Investigation, Visualization, Methodology; Matthias Teuscher, Data curation, Investigation, Writing—review and editing; Philippe Valenti, Investigation; Abderrahman Khila, Conceptualization, Data curation, Formal analysis, Supervision, Funding acquisition, Validation, Methodology, Writing—original draft, Project administration, Writing—review and editing; Martin Klingler, Conceptualization, Formal analysis, Supervision, Funding acquisition, Investigation, Writing—original draft, Project administration, Writing—review and editing; François Payre, Conceptualization, Formal analysis, Supervision, Investigation, Visualization, Writing—original draft, Project administration, Writing—review and editing

### Author ORCIDs

Miriam I Rosenberg http://orcid.org/0000-0001-5348-8247
Tzach Auman http://orcid.org/0000-0002-2233-4234
Matthias Teuscher https://orcid.org/0000-0003-2340-5716
Abderrahman Khila http://orcid.org/0000-0003-0908-483X
Martin Klingler http://orcid.org/0000-0001-8859-1965
François Payre https://orcid.org/0000-0002-8144-6711

### Decision letter and Author response

Decision letter https://doi.org/10.7554/eLife.39748.053
Author response https://doi.org/10.7554/eLife.39748.054

## Additional files

### Supplementary files

• Source data 1. Source data for charts in *Figures 1G* and *6G*, and *Figure 4—figure supplement 1–3*

DOI: https://doi.org/10.7554/eLife.39748.031

• Supplementary file 1. Supplementary information. (**A**) Genes identified in the genome wide iBeetle RNAi screen that had phenotypes resembling those of *mlpt*. iB 00966 and 09278, had the most highly penetrant RNAi phenotype with the strongest resemblance to those of *mlpt*. NCBI annotates these as belonging to a single locus LOC657900 encoding a 6592 bp mRNA (accession number XM_964327) that corresponds to *Tc-ubr3*. (**B**) Oligonucleotides used in the manuscript. Nucleotides shown in red indicate tags of parts of T7 (3' primer) and SP6 (5' primer) promoter sequences attached to gene-specific sequences for nested PCR. Nucleotides shown in blue highlight minimal T7 promoter used in subsequent in vitro T7 RNA polymerase transcription. (**C**) Oligos used for generating *Tc-svb* CRISPR mutant. (**D**) Plasmids used for generating *Tc-svb* CRISPR mutant.

DOI: https://doi.org/10.7554/eLife.39748.032

• Transparent reporting form

DOI: https://doi.org/10.7554/eLife.39748.033

## Data availability

All sequence data generated or analysed during this study are included in the manuscript and supporting files. Sequence files have been deposited in GenBank with accession numbers as follows: MG913606, MH011417, MH011418, MH181829, MH181831, MH181828, MH181830, MH181832, MH181827, MH699965. Data from iBeetle screen have already been published, and are available as indicated in the original publication: The iBeetle large-scale RNAi screen reveals gene functions for insect development and physiology. Schmitt-Engel C, Schultheis D, Schwirz J, Ströhlein N, Troelenberg N, Majumdar U, Dao VA, Grossmann D, Richter T, Tech M, Dönitz J, Gerischer L, Theis M, Schild I, Trauner J, Koniszewski ND, Küster E, Kittelmann S, Hu Y, Lehmann S, Siemanowski J, Ulrich J, Panfilio KA, Schröder R, Morgenstern B, Stanke M, Buchhholz F, Frasch M, Roth S, Wimmer EA, Schoppmeier M, Klingler M, Bucher G. Nat Commun. 2015 Jul 28;6:7822. doi: 10.1038/ncomms8822

The following datasets were generated:

| Author(s) | Year | Dataset title | Dataset URL | Database and Identifier |
|---|---|---|---|---|
| Suparna Ray | 2018 | Tribolium castaneum strain pBA19 shavenbaby (svb) mRNA, partial cds | https://www.ncbi.nlm.nih.gov/nuccore/MG913606 | NCBI GenBank, MG913606 |
| Toubiana W, Decaras A, Khila A | 2018 | Gerris buenoi shavenbaby mRNA, complete cds | https://www.ncbi.nlm.nih.gov/nuccore/MH011417 | NCBI GenBank, MH011417 |
| William T, Decaras A, Khila A | 2018 | Gerris buenoi E3 ubiquitin-protein ligase mRNA, complete cds | https://www.ncbi.nlm.nih.gov/nuccore/MH011418 | NCBI Genbank, MH011418 |
| Rosenberg MI, Ray S, Chanut-Delalande H, Decaras A, Schwertner B, Toubiana W, Auman T | 2019 | Oncopeltus fasciatus ubiquitin protein ligase E3 (UBR3) mRNA, partial CDs | https://www.ncbi.nlm.nih.gov/nuccore/MH181827 | NCBI Genbank, MH181827 |
| Rosenberg MI, Ray S, Chanut-Delalande H, Decaras A, Schwertner B, Toubiana W, Auman T, Schnellhammer I, Teuscher M, Valenti P, Khila A, Klingler M, Payre F | 2019 | Nasonia vitripennis ubiquitin protein ligase E3 (UBR3) mRNA, partial cds. | https://www.ncbi.nlm.nih.gov/nuccore/MH181828 | NCBI Genbank, MH181828 |
| Rosenberg MI, Ray S, Chanut-Delalande H, Decaras A, Schwertner B, Toubiana W, Auman T, Schnellhammer I, Teuscher M, Valenti P, Khila A, Klingler M, Payre F | 2019 | Nasonia vitripennis millepattes peptide 1, millepattes peptide 2, millepattes peptide 3, millepattes peptide 4, and millepattes peptide 5 mRNAs, complete cds. | https://www.ncbi.nlm.nih.gov/nuccore/MH181829 | NCBI Genbank, MH181829 |
| Rosenberg MI, Ray S, Chanut-Delalande H, Decaras A | 2019 | Oncopeltus fasciatus millepattes peptide 1 and millepattes peptide 2 mRNAs, complete cds. | https://www.ncbi.nlm.nih.gov/nuccore/MH181830 | NCBI Genbank, MH181830 |
| Rosenberg MI, Ray S, Chanut-Delalande H, Decaras A, Schwertner B, | 2019 | Oncopeltus fasciatus shavenbaby mRNA, partial cds. | https://www.ncbi.nlm.nih.gov/nuccore/MH181831 | NCBI Genbank, MH181831 |

Toubiana W, Auman T, Schnellhammer I, Teuscher M, Valenti P, Khila A, Klingler M, Payre F

Rosenberg MI, Ray S, Chanut-Delalande H, Decaras A, Schwertner B, Toubiana W, Auman T, Schnellhammer I, Teuscher M    2019    Gerris buenoi millepattes    https://www.ncbi.nlm.nih.gov/nuccore/MH699965    NCBI Genbank, MH699965

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
