## [Decision Letter]

[**Editorial note:** This article has been through an editorial process in which the authors decide how to respond to the issues raised during peer review. The Reviewing Editor's assessment is that all the issues have been addressed.]

Thank you for submitting your article "*millepattes* micropeptides are an ancient developmental switch required for embryonic patterning" for consideration by *eLife*. Your article has been reviewed by three peer reviewers, and the evaluation has been overseen by Diethard Tautz as the Reviewing Editor and Senior Editor. The reviewers have opted to remain anonymous.

The Reviewing Editor has highlighted the concerns that require revision and/or responses, and we have included the separate reviews below for your consideration. If you have any questions, please do not hesitate to contact us.

Summary:

The authors address the question of a conservation pattern of a segmentation gene network in insects. They ask why the gene *millepattes* that was found to be relevant for the segmentation in beetles has a different function in *Drosophila*. By studying the interaction partners of the genes in a range of insects, they suggest that the interacting network is broadly conserved, but the expression during the segmentation process has changed in *Drosophila*. Remarkably, by artificially reconstituting this expression, they can show that the network is also active in *Drosophila* at this stage. Further, the study is special interest, since the gene *millepattes* codes for short peptides only.

Major concerns:

The reviewers have identified in their reviews below, as well as in the subsequent discussion among them two major points:

1) There is a general need for better presentation of the results. The phenotypic presentations partly lack scientific rigor, and would need some sort of quantifications (even if it is only counting legs or measuring posterior segment length). Showing an image of a single example deformed embryo, and comparing different deformed embryos and claiming that the deformations are 'similar' is not sufficient. Ideally one would want to see multiple embryos from each class quantified to be able to decide to what extent the phenotypes are similar/what the spread of phenotypic severities is, etc.

2) The claim for a "reawakening" is not yet convincingly demonstrated. It needs better documentation and at least one additional experiment, as pointed out by reviewers 1 and 3. If this cannot be provided, the authors may need to drop it from the manuscript, although this would somehow limit its impact. Hence, we encourage the authors to obtain better evidence for this.

Further, the reviewers address several points where the manuscript can be improved – please check this carefully. Also, there is the unfortunate situation that the gene has three names: *mlpt, tal* and *pri*. The latter two refer both to *Drosophila* phenotypes but FlyBase lists only *tal*, i.e. there is no evidence that *pri* is a valid name. The priority for the molecular description would be *mlpt*, but the genetic description of *tal* is older. Hence, these two could be used, but *pri* should only be used in the Introduction to explain the background.

Separate reviews (please respond to each point):

*Reviewer #1:*

For summary, see above.

The study is well presented and I have only a number of minor comments, but also one major set of comments:

Major comment:

Experiments showing the re-awakening of the network in the *Drosophila* blastoderm:

1) The experiments are poorly described in the Materials and methods section – this needs to be presented much more clearly.

2) Why is Figure 5A presented as a drawing, rather than with real stainings? And why is it not described in the text?

3) Evidently, to show that the segmentation effect is dependent on *mlpt*, the Svb over-expression should also be done in a *mlpt* mutant background. Why was this not done?

Minor comments:

The first part of the Introduction is a bit confusing and should be re-written: it starts with describing the phenomenon of small open reading frames, but cites then the work of Payre et al., which describes a different gene. Also, while explaining that three papers describe the *millepattes/tarsalless*/polished rice gene, the authors use then the term "Pri peptides", which is confusing. This happens also at other places (e.g. Figure 5) – a consistent use of the names is required. Given that *millepattes* is used in the title and since it was the first name published, it should be used throughout.

Figure 4I shows a larva, not a fly (correct text).

Discussion: the effect on segmentation is discussed, but leaves out the question why transformations of segment identity can be observed.

*Reviewer #2:*

This manuscript reports expression and functional analysis of *mlpt*, svb, and *ubr3* in a range of insects. This work shows that the functional interactions of the products of these three genes is conserved in insects and likely regulated cuticle differentiation, leg development and segmentation ancestrally in insects although the function in segmentation has been lost in dipterans.

This is a very thorough and rigorous study with impressive functional data from many insects. The results are also very interesting because they not only show the functional conservation of this module, but provide new insights into how is regulates different developmental processes. It is also very interesting how the spatial and temporal expression of the three genes is a key aspect of the modulation of their functional interactions.

While I have no substantive concerns I think the manuscript would be improved by the authors addressing the following two points:

1) The authors show that the functional interaction between mlpt, *ubr3* and *svb* can be ectopically induced in early *Drosophila* embryos by over expressing *svb* – however this is not really 'reawakening' of the role of this module in segmentation as suggested by the authors because it leads to segmentation defects – not the proposed ancestral role of this module during segmentation – therefore I think this needs to be toned down and rephrased.

2) The first part of the Results section describing the function and expression of *svb* and *ubr3* in *Tribolium* is very difficult to follow and needs to be rewritten before publication – it jumps around too much between embryos and legs and from gene to gene and is very hard to read. I also suggest reordering this section to first describe the express patterns of the genes in *Tribolium* and then go on to describe the effects of the RNAi and CRISPR – I found it quite odd to first read about the function and then only afterwards about the expression of the genes – which is essential to understand the functional tests.

Minor comments:

The third sentence of the Discussion ('We surmise…") is very confusing – please clarify what is meant by this with respect to the evolution of different modes of segmentation.

*Reviewer #3:*

In this manuscript entitled "*millepattes* micropeptides are an ancient developmental switch required for embryonic patterning" Ray and Rosenberg and colleagues investigate the conservation of the gene regulatory module comprised of the short peptides *Pri/Mlpt/Tarsalless, Svb* and *Ubr3* in 4 different insect species. The lab had previously identified this gene module in *Drosophila* where it is required for trichome formation in the larval cuticle (Zanet et al., 2015). Given that the Pri/Mlpt peptides are highly conserved in insects, yet loss of function phenotypes differ between *Drosophila* and beetle), the authors set out to address two main questions:

1) Conservation of the gene regulatory module identified in *Drosophila* in other insect species.

2) 'Functionality' of the gene regulatory module in early segmentation/patterning also in *Drosophila*, in which it is not required for early embryonic patterning.

The authors' main conclusions are that the gene regulatory module comprised of Pri/Mlpt/Tarsalless, Svb and Ubr3 is conserved in *Tribolium* and 2 further insect species. Furthermore, the authors employ Svb overexpression in the early *Drosophila* embryo to 'reawaken' the ancestral developmental switch during early *Drosophila* embryogenesis. The embryonic defects upon Svb and OvoB (the activated form of Svb) overexpression are interpreted as evidence that this pathway can be 're-awakened' in the early *Drosophila* embryo, but is normally not active since Svb is not expressed in the early *Drosophila* embryo.

The main strength of the paper is that it investigates the interesting open question why the loss of function phenotypes in *Drosophila* and *Tribolium* are so different despite the fact that the Pri/Mlpt/Tarsalless peptides are highly conserved. Two possibilities why this could be the case are 1) due to different downstream pathways that the peptides may regulate in different insect species; 2) different expression patterns/timing of the same pathway components in the different species. The authors report here that the downstream pathway components (the 'gene module') are conserved across species, and that different timings and patterns of expression are the main reason behind the different phenotypes.

However, I have two major concerns regarding interpretation of the data. Apart from my concerns regarding soundness of the conclusions drawn based on the data presented, a major weakness of this manuscript is figure quality and style. Whenever possible I have included suggestions for improvements.

Major weaknesses:

1) The core observation, and – based on the Abstract, most interesting finding – is based on 'reawakening' of the functionality of the module in segmentation by overexpressing Svb in *Drosophila* embryos. However, I don't think one can conclude from defects observed upon OE of *svb* in *Drosophila* that this gene module can be 'reawakened' by giving back *svb* (= the functionality of the gene module (*mlpt, svb, ubr3*)). In the other species apart from *Drosophila, svb* is expressed and needed to form a normally patterned embryo, and LOSS of its function leads to defects. However, the authors show here that the reverse (= OE of *svb* in the entire embryo) causes defects in *Drosophila*. The defects observed appear unrelated to the observed patterning defects caused by loss of the gene module in the other 3 species where loss of function causes posterior truncation + extra pair of legs, while *svb* OE causes a deformed embryo.

Suggestion 1: Essential experiment to link the *svb* OE phenotype to *pri* (and with this to the gene module): OE of *svb* in the context of a *pri* mutant should not cause any defects (or different defects if repressor function may cause independently phenotypes, which seems unlikely given that OvoB causes similar (?) OE phenotypes as *svb*) if the active form of *svb* in the embryo is indeed causing these phenotypes- is the case? However, this experiment will in my opinion still not allow the authors to conclude that the defects upon OE of *svb* in *Drosophila* are related to the functionality of the gene module in the other species in segmentation. The only way to really show that this gene module – if present/expressed – can also be functional in segmentation in *Drosophila* would in my opinion be to attempt to 'rescue' a segmentation mutant (which is very unlikely to work for multiple reasons, foremost since *Drosophila* seems to have evolved a different way of segmenting the early embryo, and this module thus became dispensable over evolutionary time in this species).

Suggestion 2: OE would always need a proper negative control since OE in general can cause defects that are unrelated to normal protein function.

2) Problem with assessing 'similarity' of described phenotypes

– Identification of Svb/Ubr3 module also in *tribolium* -> kd of these factors causes 'similar' pts; it's hard to judge from the images how similar the phenotypes really are (e.g. whether there are more legs also in the *svb* knockdown; labelling of the legs, and better-quality figures are absolutely needed to assess whether the phenotypes are indeed 'similar').

– *svb* mutant phenotype looks – based on the image provided (is this Figure 1F? the reference to this figure in the text is unclear with Figure 1 and 4) – very different than the *mlpt* mutant (multiple legs; truncated abdomen) vs *svb* mutant phenotype (abdominal truncation not clear; very short legs) – compare 1B with 1F.

– I also have problems in seeing the phenotypic similarity in the other two species (Figure 3). The phenotypes in these species appear very different to *mlpt* phenotype in Tribolium, whose most characteristic feature is more legs and shorter abdomen; none of these phenotypes shown here appear to have more legs? Rather posterior deletion and fusion of thoracic segments? I do not recognize the similarity; more convincing data + better images + quantification is needed. Also, it would be essential to show convincing data for segmentation phenotypes (as opposed to a deformed embryo); this could for example be done in a similar way as for *Tribolium* (with wg probe; Figure 1H).

Apart from my two major concerns regarding the interpretation of the data, I have four general points that should be addressed to improve the quality and readability of the manuscript:

– Figures are of very poor quality in the uploaded pdf, and the figure legends barely or not at all readable.

– Phenotypes would need to be quantified in all the figures (e.g. numbers of legs, length abdomen and thorax…).

– Refer to figures in the order mentioned (do not start in Introduction with Figure 5).

– Mention all three gene names (polished rice/mlpt/tarsalless) when first introducing them (already in Abstract), and then stick to one name (or at least, 1 name per species).

Further minor points are listed below:

– Knockdown of *mlpt* causing upregulation of *svb* also in *Tribolium* (Supplementary figure 1 F-O): why is there upregulation throughout the whole embryo/larvae, and not only in the *mlpt* expression domains? (as shown in Figure 1I-M)?

– I think the in situ images from Dros *svb* and *pri* (Figure 4) should go together (as comparison) with the in situ images of *Tribolium svb* and *mlpt* (Figure 1 1-M') since this is a very important point that the *svb* expression in Dros is absent early.

– Absence of colocalization in Figure 4: this is hard to judge from single in situs – double in situs (between *svb* and *mlpt*) would be needed to judge whether the bands are overlapping or not (and at least in Figure C/C' and D/D').

– I think one cannot suggest similar molecular interactions based on "the similarities in transcript, and protein structure and disorder disposition profiles between *Drosophila* and Tribolium" – there are plenty of proteins that share a common structure/domain (transcript should be irrelevant for molecular interactions on protein-level) but have very different interaction partners that differ based on the amino acid sequence. What would in my opinion be better/more correct to stress is that these are homologs (or orthologs if this is known); orthology would suggest that they share common interaction partners.

– Description of expression domains is confusing – initially stated that *svb* expressed throughout embryogenesis in Dros, but then a bit later on 'absence of *svb* expression in the early fly embryo"; in general: descriptions of expression patterns are too lengthy, and would need to be compressed to the most essential key points; only focus on essential things.

– The wording of the gene expression paragraph about the other species reads as if the authors would use the expression domains as evidence for function (while later on they do RNAi). I would suggest two possibilities to make this clearer: 1) start with the most convincing data (functional data, RNAi) and just mention in Suppl or as a confirmation that the gene expression was consistent with the phenotype; or 2) say first that to see whether those genes were expressed at all at the right time to potentially have a function in segmentation, gene expression patterns were analyzed. Then functional data to get evidence for functionality.

– What would be interesting to discuss more is how the mostly non-overlapping expression patterns of *svb* and *mlpt* in *Tribolium* could lead to such homeotic transformation phenotypes; in which way are the expression patterns consistent with the abdomen -> thorax transformation?

– Consistency with nomenclature:

a) Naming: mention the first time all 3 names, but then stick to one (and not talk in the Introduction mainly about Pri, while the Abstract does not contain that abbreviation at all).

b) Follow the nomenclature guidelines for these species to refer to genes, RNA and proteins.

Additional data files and statistical comments:

Phenotypes would need to be quantified in all the figures (e.g. numbers of legs, length abdomen and thorax…). No statistical analyses are provided in the current version of the manuscript.

---

## [Author Response]

Major concerns:The reviewers have identified in their reviews below, as well as in the subsequent discussion among them two major points1) There is a general need for better presentation of the results. The phenotypic presentations partly lack scientific rigor, and would need some sort of quantifications (even if it is only counting legs or measuring posterior segment length). Showing an image of a single example deformed embryo, and comparing different deformed embryos and claiming that the deformations are 'similar' is not sufficient. Ideally one would want to see multiple embryos from each class quantified to be able to decide to what extent the phenotypes are similar/what the spread of phenotypic severities is, etc.

We collected new data to better quantify and compare the various phenotypes across genes and species. The revised version now incorporates these new data, including several pictures to illustrate each phenotype, quantitative measurement and statistical tests.

2) The claim for a "reawakening" is not yet convincingly demonstrated. It needs better documentation and at least one additional experiment, as pointed out by reviewers 1 and 3. If this cannot be provided, the authors may need to drop it from the manuscript, although this would somehow limit its impact. Hence, we encourage the authors to obtain better evidence for this.

We now provide complementary lines of evidence demonstrating functional interaction between Svb, Tal and Ubr3 and the effect of this interaction on *Drosophila* segmentation. The additional experiment suggested by the reviewers bore the expected result and has been included (see below).

Further, the reviewers address several points where the manuscript can be improved – please check this carefully. Also, there is the unfortunate situation that the gene has three names: mlpt, tal and pri. The latter two refer both to Drosophila phenotypes but FlyBase lists only tal, i.e. there is no evidence that pri is a valid name. The priority for the molecular description would be mlpt, but the genetic description of tal is older. Hence, these two could be used, but pri should only be used in the Introduction to explain the background.

The nomenclature has been simplified, according to these remarks.

Separate reviews (please respond to each point):

Reviewer #1:

For summary, see above.The study is well presented and I have only a number of minor comments, but also one major set of comments:Major comment:Experiments showing the re-awakening of the network in the Drosophila blastoderm:1) The experiments are poorly described in the Materials and methods section – this needs to be presented much more clearly.

The revised version clarifies this point and gives better description of former and new experiments, in both main text, figure legends and Materials and methods.

2) Why is Figure 5A presented as a drawing, rather than with real stainings? And why is it not described in the text?

As suggested by the reviewer, the revised figure (now Figure 6) presents actual staining that shows the expression of *svb* and *tal* mRNAs.

3) Evidently, to show that the segmentation effect is dependent on mlpt, the Svb over-expression should also be done in a mlpt mutant background. Why was this not done?

The revised version now addresses this important point and shows that the lack of *tal/mlpt* function abrogates the segmentation defects resulting from *svb* ectopic expression in early embryos (Figure 6 E-F’, G). This conclusion is further reinforced by a set of independent experiments, making use of two engineered forms of the Svb protein, one mimicking constitutive *mlpt*-mediated processing and the other that bears 3 point mutations preventing Ubr3 binding/ubiquitinylation (Figure 6—figure supplement 2). We find that while the former induces strong segmentation defects, the latter does not affect segmentation (Figure 6 A-D’’’, G). Collectively, these new pieces of evidence provide strong support to the conclusion that the ectopic function of Svb during *Drosophila* segmentation relies on Pri and Ubr3.

Minor comments:The first part of the Introduction is a bit confusing and should be re-written: it starts with describing the phenomenon of small open reading frames, but cites then the work of Payre et al., which describes a different gene.

The Introduction section has been entirely rewritten taking into account these comments.

Also, while explaining that three papers describe the millepattes/tarsalless/polished rice gene, the authors use then the term "Pri peptides", which is confusing. This happens also at other places (e.g. Figure 5) – a consistent use of the names is required. Given that millepattes is used in the title and since it was the first name published, it should be used throughout.

The existence of different names for a same gene within and across species is always confusing. As suggested by the editor, we thus stick to *mlpt* in *Tribolium* and all other species we studied, and keep *tal* only for *Drosophila.*

Figure 4I shows a larva, not a fly (correct text).

The typo has been corrected.

Discussion: the effect on segmentation is discussed, but leaves out the question why transformations of segment identity can be observed.

The discussion has been extensively edited in the revised version.

Reviewer #2:

This manuscript reports expression and functional analysis of mlpt, svb, and ubr3 in a range of insects. This work shows that the functional interactions of the products of these three genes is conserved in insects and likely regulated cuticle differentiation, leg development and segmentation ancestrally in insects although the function in segmentation has been lost in dipterans.This is a very thorough and rigorous study with impressive functional data from many insects. The results are also very interesting because they not only show the functional conservation of this module, but provide new insights into how is regulates different developmental processes. It is also very interesting how the spatial and temporal expression of the three genes is a key aspect of the modulation of their functional interactions.While I have no substantive concerns I think the manuscript would be improved by the authors addressing the following two points:1) The authors show that the functional interaction between mlpt, ubr3 and svb can be ectopically induced in early Drosophila embryos by over expressing svb – however this is not really 'reawakening' of the role of this module in segmentation as suggested by the authors because it leads to segmentation defects – not the proposed ancestral role of this module during segmentation – therefore I think this needs to be toned down and rephrased.

We agree with the reviewer on this point. The Abstract and text has been rewritten to take into account this point.

2) The first part of the Results section describing the function and expression of svb and ubr3 in Tribolium is very difficult to follow and needs to be rewritten before publication – it jumps around too much between embryos and legs and from gene to gene and is very hard to read. I also suggest reordering this section to first describe the express patterns of the genes in Tribolium and then go on to describe the effects of the RNAi and CRISPR – I found it quite odd to first read about the function and then only afterwards about the expression of the genes – which is essential to understand the functional tests.

We agree that the former text was probably too detailed and thus difficult to follow. We have entirely rewritten and simplified this section, referring to a new synthetic table that compares the various phenotypical aspects of RNAi and CRISP animals (Table 1). We still believed that it is important to start from results from the large-scale iBeetle screen (Donitz et al., 2015), which provides an unbiased path for *Ubr3* identification. This strategy, we think, would give the context that led us to formulate the hypotheses leading to the experiments we conducted.

Minor comments:The third sentence of the Discussion ('We surmise…") is very confusing – please clarify what is meant by this with respect to the evolution of different modes of segmentation.

The Discussion section has been rewritten and includes in depth editing.

Reviewer #3:

[…] The main strength of the paper is that it investigates the interesting open question why the loss of function phenotypes in Drosophila and Tribolium are so different despite the fact that the Pri/Mlpt/Tarsalless peptides are highly conserved. Two possibilities why this could be the case are 1) due to different downstream pathways that the peptides may regulate in different insect species; 2) different expression patterns/timing of the same pathway components in the different species. The authors report here that the downstream pathway components (the 'gene module') are conserved across species, and that different timings and patterns of expression are the main reason behind the different phenotypes.However, I have two major concerns regarding interpretation of the data. Apart from my concerns regarding soundness of the conclusions drawn based on the data presented, a major weakness of this manuscript is figure quality and style. Whenever possible I have included suggestions for improvements.Major weaknesses:1) The core observation, and – based on the Abstract, most interesting finding – is based on 'reawakening' of the functionality of the module in segmentation by overexpressing Svb in Drosophila embryos. However, I don't think one can conclude from defects observed upon OE of svb in Drosophila that this gene module can be 'reawakened' by giving back svb (= the functionality of the gene module (mlpt, svb, ubr3)). In the other species apart from Drosophila, svb is expressed and needed to form a normally patterned embryo, and LOSS of its function leads to defects. However, the authors show here that the reverse (= OE of svb in the entire embryo) causes defects in Drosophila. The defects observed appear unrelated to the observed patterning defects caused by loss of the gene module in the other 3 species where loss of function causes posterior truncation + extra pair of legs, while svb OE causes a deformed embryo.Suggestion 1: Essential experiment to link the svb OE phenotype to pri (and with this to the gene module): OE of svb in the context of a pri mutant should not cause any defects (or different defects if repressor function may cause independently phenotypes, which seems unlikely given that OvoB causes similar (?) OE phenotypes as svb) if the active form of svb in the embryo is indeed causing these phenotypes- is the case? However, this experiment will in my opinion still not allow the authors to conclude that the defects upon OE of svb in Drosophila are related to the functionality of the gene module in the other species in segmentation. The only way to really show that this gene module – if present/expressed – can also be functional in segmentation in Drosophila would in my opinion be to attempt to 'rescue' a segmentation mutant (which is very unlikely to work for multiple reasons, foremost since Drosophila seems to have evolved a different way of segmenting the early embryo, and this module thus became dispensable over evolutionary time in this species).

As suggested, we did assay for the requirement of *tal/mlpt* and *ubr3* function for the effects of Svb ectopic expression on embryo segmentation, using two complementary approaches (see new Figure 6). As expected by the reviewer, removing *tal* function –or preventing Ubr3 interaction- both abrogates segmentation defects upon Svb expression. While these new results unambiguously show the role of all three players in this phenotype, we acknowledge that a formal demonstration of reawaking ancestral function in segmentation remains currently out of reach. We rephrase all concerned sections to take into account this point.

Suggestion 2: OE would always need a proper negative control since OE in general can cause defects that are unrelated to normal protein function.

In addition to GFP, that the full Svb protein (1354aa) that bears three point mutations substituting N-term K->A is devoid of activity in segmentation (but still acts during later epidermal differentiation), now provides an interesting and relevant control (Figure 6 and Figure 6—figure supplement 2).

2) Problem with assessing 'similarity' of described phenotypes– Identification of Svb/Ubr3 module also in tribolium -> kd of these factors causes 'similar' pts; it's hard to judge from the images how similar the phenotypes really are (e.g. whether there are more legs also in the svb knockdown; labelling of the legs, and better-quality figures are absolutely needed to assess whether the phenotypes are indeed 'similar').

We now include a formal quantification of the defects across genes and species, now presented in new Figure 1, Figure 6 and Figure 4—figure supplements 1,2,3. We also provide better quality pics and more explicit labelling. In addition, we provide four additional supplementary figures (Figure 1—figure supplement 1-4) and a summary table (Table 1) to help comparing the RNAi and CRISPR phenotypes.

– svb mutant phenotype looks – based on the image provided (is this Figure 1F? the reference to this figure in the text is unclear with Figure 1 and 4) – very different than the mlpt mutant (multiple legs; truncated abdomen) vs svb mutant phenotype (abdominal truncation not clear; very short legs) – compare 1B with 1F.

As we now mention in the supplemental section (Figure 1—figure supplement 4), the *Tc-svb* segmentation phenotype is indeed somewhat weaker than in the RNAi knockdown. This likely is due to a maternal contribution to *Tc-svb* expression in the embryo, as we will describe in more detail in a follow-up paper (Ray and Klingler, in preparation).

– I also have problems in seeing the phenotypic similarity in the other two species (Figure 3). The phenotypes in these species appear very different to mlpt phenotype in Tribolium, whose most characteristic feature is more legs and shorter abdomen; none of these phenotypes shown here appear to have more legs? Rather posterior deletion and fusion of thoracic segments? I do not recognize the similarity; more convincing data + better images + quantification is needed. Also, it would be essential to show convincing data for segmentation phenotypes (as opposed to a deformed embryo); this could for example be done in a similar way as for Tribolium (with wg probe; Figure 1H).

We provide additional data and quantification that better illustrate the phenotype similarities within each species (Figure 1 G; Figure 1—figure supplements 1-4; Figure 4—figure supplements 1,2,3; Figure 5—figure supplement 5 and Table 1). We agree with the reviewer that transformation of abdominal toward thoracic identity appears so far to be limited to *Tribolium*, since all other species display posterior truncation and fused segments, often of uncertain identity. It is interesting to note that, in *Tribolium,* knockdown of other gap genes has been reported to cause homeotic phenotypes including *Tc-Hunchback* (Marques-Souza et al., 2008), *Tc-Giant* (Bucher and Klingler, 2004) and *Tc-Kruppel* (Cerny et al., 2005), as now mentioned in the revised manuscript (see Introduction).

As requested, we also provide new data to document En/Wg and homeotic gene expression in additional species (Figure 4—figure supplements 4,5,6; Figure 5—figure supplement 1; Figure 6—figure supplement 1).

Apart from my two major concerns regarding the interpretation of the data, I have four general points that should be addressed to improve the quality and readability of the manuscript:– Figures are of very poor quality in the uploaded pdf, and the figure legends barely or not at all readable.

We apologize for what appears to have been a technical problem; the new version definitively solves these issues.

– Phenotypes would need to be quantified in all the figures (e.g. numbers of legs, length abdomen and thorax…).

Quantitative analyses have been added to Figure 1; Figure 6; Figure 4—figure supplement 1,2,3 as well as in Table 1. Since outside of *Tribolium* there are no homeotic transformations leading to additional leg bearing segments, we have quantitated the conserved phenotypic features, most importantly, leg shortening and abdominal shortening.

– Refer to figures in the order mentioned (do not start in Introduction with Figure 5).

This has been corrected.

– Mention all three gene names (polished rice/mlpt/tarsalless) when first introducing them (already in Abstract), and then stick to one name (or at least, 1 name per species).

As also requested by other reviewers and the editor, we homogenize the nomenclature throughout.

Further minor points are listed below:– Knockdown of mlpt causing upregulation of svb also in Tribolium (Supplementary figure 1 F-O): why is there upregulation throughout the whole embryo/larvae, and not only in the mlpt expression domains? (as shown in Figure 1I-M)?

We agree that these data were not entirely convincing and may suffer of technical limitations, since they rely on detecting a specific *Tc-Svb* isoform making use of an antibody raised against the *Drosophila* protein. Hence, we preferred removing this figure, since the mutual regulation between *svb* and *mlpt* is well demonstrated using validated *in situ* probes, in both *Tribolium* and *Oncopeltus* (Figure 2—figure supplement 2).

– I think the in situ images from Dros svb and pri (Figure 4) should go together (as comparison) with the in situ images of Tribolium svb and mlpt (Figure 1 1-M') since this is a very important point that the svb expression in Dros is absent early.

We understand this concern, which was clearly meaningful given the mention of *svb* expression in flies from the former introduction. We hope that this point is no longer relevant in our deeply rewritten revised version, and believe that it makes sense to compare *svb* and *mlpt/tal* expression between the two long germand species *Drosophila* and *Nasonia*.

– Absence of colocalization in Figure 4: this is hard to judge from single in situs – double in situs (between svb and mlpt) would be needed to judge whether the bands are overlapping or not (and at least in Figure C/C' and D/D').

Rephrased.

– I think one cannot suggest similar molecular interactions based on "the similarities in transcript, and protein structure and disorder disposition profiles between Drosophila and Tribolium" – there are plenty of proteins that share a common structure/domain (transcript should be irrelevant for molecular interactions on protein-level) but have very different interaction partners that differ based on the amino acid sequence. What would in my opinion be better/more correct to stress is that these are homologs (or orthologs if this is known); orthology would suggest that they share common interaction partners.

We agree that this sentence was confusing. Svb proteins across insect species are clearly homologs, and there is a single corresponding gene in each species. A significant hint of the evolutionary conservation of Svb/Ubr3 interaction is, as expected, a strong conservation of the amino acid sequence of the interaction domain previously reduced down to 31 aa (Zanet et al., 2015), including the 3 lysine residues ubiquitinated by Ubr3 (and now shown to be required for segmental function in flies, see Figure 6). Conservation of this motif appears an even stronger argument, given the rapidly-evolving nature of the remaining protein sequence, except the DNA-binding domain. The same is also true for the profile of predicted disorder disposition, across all insect species. We redrew the figure (Figure 1—figure supplement 5) and rephrase the whole paragraph to explicitly highlight these data.

– Description of expression domains is confusing – initially stated that svb expressed throughout embryogenesis in Dros, but then a bit later on 'absence of svb expression in the early fly embryo"; in general: descriptions of expression patterns are too lengthy, and would need to be compressed to the most essential key points; only focus on essential things.

The text has been extensively rephrased accordingly.

– The wording of the gene expression paragraph about the other species reads as if the authors would use the expression domains as evidence for function (while later on they do RNAi). I would suggest two possibilities to make this clearer: 1) start with the most convincing data (functional data, RNAi) and just mention in Suppl or as a confirmation that the gene expression was consistent with the phenotype; or 2) say first that to see whether those genes were expressed at all at the right time to potentially have a function in segmentation, gene expression patterns were analyzed. Then functional data to get evidence for functionality.

Rephrased.

– What would be interesting to discuss more is how the mostly non-overlapping expression patterns of svb and mlpt in Tribolium could lead to such homeotic transformation phenotypes; in which way are the expression patterns consistent with the abdomen -> thorax transformation?

The homeotic transformations observed in *Tribolium* are consistent with Mlpt function (presumably in conjunction with Svb) as a gap gene. Other gap genes in *Tribolium* also give homeotic transformation phenotypes in RNAi or mutant backgrounds (see above).

Given that *Tc-svb* and *mlpt* are coexpressed in the blastoderm and early germ rudiment, regulation of Svb could occur at this stage and influence subsequent segmentation and Hox gene regulation. Alternatively, diffusion of Mlpt peptides at stages when their expression domains are complementary could result in activation of Svb. Without available tools for protein/peptide localization, these possibilities cannot be distinguished at this time. We do know, however, that in *Tribolium* a loss of anterior AbdA and Ubx expression is observed, which explains the transformation; these data will appear in a separate manuscript describing Hox regulation by a *hs-mlpt* transgene in *Tribolium* (Ray and Klingler, in preparation).

– Consistency with nomenclature:a) Naming: mention the first time all 3 names, but then stick to one (and not talk in the Introduction mainly about Pri, while the Abstract does not contain that abbreviation at all).

This has been corrected.

b) Follow the nomenclature guidelines for these species to refer to genes, RNA and proteins.

This has been corrected.

Additional data files and statistical comments:Phenotypes would need to be quantified in all the figures (e.g. numbers of legs, length abdomen and thorax…). No statistical analyses are provided in the current version of the manuscript.

This has been done (Figure 1; Figure 4—figure supplements 1,2,3; Figure 6).